# A neurobiological association of revenge propensity during intergroup conflict

Xiaochun Han[1], Michele J Gelfand[2], Bing Wu[3], Ting Zhang[1], Wenxin Li[4], Tianyu Gao[1], Chenyu Pang[1], Taoyu Wu[1], Yuqing Zhou[1], Shuai Zhou[3], Xinhuai Wu[3]*, Shihui Han[1]*

[1]School of Psychological and Cognitive Sciences, PKU-IDG/McGovern Institute for Brain Research, Beijing Key Laboratory of Behavior and Mental Health, Peking University, Beijing, China; [2]Department of Psychology, University of Maryland, College Park, United States; [3]Department of Radiology, The 7th Medical Center of PLA General Hospital, Beijing, China; [4]Peking-Tsinghua Center for Life Sciences, Peking University, Beijing, China

**Abstract** Revenge during intergroup conflict is a human universal, but its neurobiological underpinnings remain unclear. We address this by integrating functional MRI and measurements of endogenous oxytocin in participants who view an ingroup and an outgroup member's suffering that is caused mutually (Revenge group) or by a computer (Control group). We show that intergroup conflict encountered by the Revenge group is associated with an increased level of oxytocin in saliva compared to that in the Control group. Furthermore, the medial prefrontal activity in response to ingroup pain in the Revenge group but not in the Control group mediates the association between endogenous oxytocin and the propensity to give painful electric shocks to outgroup members, regardless of whether they were directly involved in the conflict. Our findings highlight an important neurobiological correlate of revenge propensity, which may be implicated in conflict contagion across individuals in the context of intergroup conflict.

**\*For correspondence:**
bei925@sina.com (XW);
shan@pku.edu.cn (SH)

**Competing interests:** The authors declare that no competing interests exist.

## Introduction

Revenge, which refers to taking actions of harming someone in retaliation for an injury (*Elshout et al., 2015*; *Jackson et al., 2019*), is a global phenomenon and a causal factor in many homicides and transgenerational conflicts (*Kopsaj, 2016*; *Jackson et al., 2019*). Although revenge is an aggressive act, not all aggressive acts represent vengeance. For example, unsolicited acts of aggression, like deviance, incivility, and bullying, would not count as revenge (*Raver and Barling, 2008*; *Jackson et al., 2019*). Revenge often occurs between families or clans when an outgroup member brings harm to an ingroup member which, in turn, induces retaliation upon outgroup members (*Ericksen and Horton, 1992*). According to early social psychological theories (*Allport et al., 1954*; *Brewer, 1999*), a desire to help the ingroup ('ingroup love') and/or an aggressive motivation to hurt the outgroup ('outgroup hate') may drive participation in intergroup conflict by taking revenge. Recent behavioral research using economic games suggests that ingroup love plays a key role in driving economic punishment towards the outgroup (*Halevy et al., 2008*; *De Dreu, 2010*; *De Dreu et al., 2010*; *Halevy et al., 2012*). Nevertheless, despite the severe social consequences of revenge, its neurobiological underpinnings remain unclear. Building upon previous findings (*Halevy et al., 2008*; *De Dreu, 2010*; *De Dreu et al., 2010*; *Halevy et al., 2012*), we suggest that there may be a neurobiological mechanism that links perceived ingroup pain caused by an outgroup and the propensity to seek revenge upon an outgroup during intergroup conflict. The present work specifically examined the hormonal (i.e., oxytocin) and neural responses to ingroup suffering caused by an outgroup that predict revenge propensity against outgroups.

Previous brain imaging research has revealed neural responses to ingroup/outgroup members' suffering, yet they have been done in contexts that lack the key character of real-life intergroup conflict, that is ingroup and outgroup members causing each other's pain. Functional magnetic resonance imaging (fMRI) studies have identified increased activity in both the empathy network (e.g., the anterior midcingulate [aMCC] and anterior insula [AI]) and the theory-of-mind network (e.g., the medial prefrontal cortex [mPFC] and the temporoparietal junction [TPJ]) in response to ingroup pain (*Hein et al., 2010*; *Cikara et al., 2011*; *Han, 2018*). Outgroup pain, on the other hand, is related to enhanced activity in the reward system (e.g., the ventral striatum and nucleus accumbens [*Hein et al., 2010*; *Cikara et al., 2011*; *Luo et al., 2015*]). In addition, the mPFC activity in response to perceived pain is associated with decisions to help ingroup members (*Hein et al., 2010*; *Mathur et al., 2010*), and the activity in the nucleus accumbens predicts decisions not to help outgroup members (*Hein et al., 2010*; *Luo et al., 2015*). These findings highlight ingroup favoritism in brain responses to others' pain as neural underpinnings of ingroup love but leave open a critical question: are brain responses to ingroup pain inflicted by outgroup members during intergroup conflict associated with subsequent revenge? Specifically, it is unclear whether activities in the empathy and/or theory-of-mind networks in response to perceived ingroup pain are associated with revenge motives during intergroup conflict. If revenge aims to bring suffering to an outgroup in order to get reward during intergroup conflict, one may expect the involvement of the reward system in decision making related to outgroup punishment (*Hein et al., 2010*; *Cikara et al., 2011*; *Han, 2018*). However, if the goal of revenge is to help ingroup members who suffer from physical harm caused by an outgroup (*Lickel et al., 2006*), the mPFC, which responds to ingroup pain and is associated with ingroup help (*Hein et al., 2010*; *Mathur et al., 2010*), may be associated with tendencies to punish the outgroup.

Previous fMRI research has also examined the neurobiological correlates of punishment decision-making pertaining to those who have violated social norms in economic games (*Seymour et al., 2007*; *Krueger and Hoffman, 2016*). Punishment decisions to prevent social norm violations have been associated with increased activities in both the empathy and theory-of-mind networks, including the aMCC, AI, and mPFC (*Krueger and Hoffman, 2016*). Yet these studies focused on brain activities related to punishment decisions rather than neurobiological mechanisms that link perceived ingroup suffering to propensity to seek revenge upon outgroups. During previously studied economic games, punishment decisions were likely motivated by prevention of social norm violations rather than by perceived physical harm to ingroup members caused by outgroup members, which characterizes most revenge behavior in real-life situations.

Finally, at the hormone level, recent research reported increased levels of urinary oxytocin (OT) — a nine-amino-acid peptide synthesized in hypothalamic cells — in chimpanzees immediately before and during border patrols and intergroup encounters (*Samuni et al., 2017*). Likewise, intranasal administration of OT (vs. placebo) in humans enhanced both empathic neural responses to ingroup pain (*Sheng et al., 2013*) and individuals' contributions to ingroup payoffs (*De Dreu et al., 2010*; *De Dreu, 2010*). OT administration also promotes motivation to sacrifice outgroup targets (*De Dreu et al., 2011*) and facilitates within-group coordination for successful outgroup attack during economic games (*Zhang et al., 2019*). These findings shed light on a functional role of the oxytocinergic system in decision making related to outgroup punishment. However, there has been little direct evidence to show that endogenous OT in humans is modulated during intergroup conflict (but see *Levy et al., 2016*). In addition, neural architectures that mediate endogenous OT and revenge propensity during intergroup conflict have been largely unexplored. Among the brain regions in which activities are sensitive to ingroup pain, the mPFC contains OT-sensitive neurons (*Ninan, 2011*). The mPFC, cingulate, and insula express OT receptors (*Gimpl and Fahrenholz, 2001*; *Macdonald and Macdonald, 2010*; *Boccia et al., 2013*) and mPFC/aMCC activities are modulated by administered OT (*Sabihi et al., 2014*; *Eckstein et al., 2015*; *Liu et al., 2017*; *Wang et al., 2017*). However, to date, whether the neural systems that are involved in empathy or theory-of-mind link endogenous OT to revenge propensity during intergroup conflict has yet to be examined.

A key challenge to our ability to address these issues empirically is the need for an experimental paradigm of intergroup conflict that can be used in a neuroimaging laboratory setting to measure: (i) neurobiological responses to perceived ingroup physical pain caused by an outgroup and (ii) revenge propensity to bring physical harm to the outgroup. Another challenge for empirical research on the neurobiological association of revenge propensity during intergroup conflict is

the need to disentangle the effect of the key component of revenge (i.e., the punishment of outgroup members for harm that they have caused to one's ingroup) from other concomitant but non-essential factors, including perceived group identity (*Kahn et al., 2017*), negative evaluation of the outgroup (*Schiller et al., 2014*), and decreased empathy for outgroup pain (*Hein et al., 2010*; *Cikara et al., 2011*; *Han, 2018*). These factors themselves may lead to negative treatment of outgroup members on the basis of ingroup biases in social behavior that occur even in the absence of intergroup conflict. It is therefore necessary to examine neurobiological responses in two conditions in which ingroup biases in emotions, attitudes, and behavior are matched, but the motive to punish the outgroup is different. That is, in the revenge condition, individuals punish outgroup members because they bring physical harm to the ingroup, whereas in the control condition, all else being equal, individuals punish outgroup members to show their ingroup favoritism in the absence of perceived intergroup conflict.

Toward these ends, we developed a new neural-behavioral paradigm that simulates real-life revenge during intergroup conflict. In this paradigm, participants viewed an ingroup and an outgroup member who gave each other painful electric shocks (Revenge group) or received electric shocks given by a computer (Control group) during a competitive game. The key difference between the two conditions is whether outgroup members brought physical harm to ingroup members during an intergroup conflict while all other aspects of the experimental manipulations were the same for the two groups. We measured participants' salivary levels of OT, brain responses to perceived ingroup pain, and revenge propensity to bring physical harm to the outgroup. These measures allowed us to investigate the neurobiological correlates of how harm to an ingroup member caused by an outgroup member inspires an uninvolved ingroup member to punish outgroup members.

On the basis of the findings of increased endogenous OT immediately before and during border patrols and intergroup encounters in chimpanzees (*Samuni et al., 2017*), we hypothesized that endogenous OT in humans is also sensitive to intergroup conflict and that salivary levels of OT would increase in the Revenge compared to Control groups. As brain regions in both the empathy network and the theory-of-mind network express OT receptors (*Gimpl and Fahrenholz, 2001*; *Macdonald and Macdonald, 2010*; *Boccia et al., 2013*), and as activities in both networks are modulated by administered OT (*Sabihi et al., 2014*; *Eckstein et al., 2015*; *Liu et al., 2017*; *Wang et al., 2017*), activities in both networks in response to ingroup members' pain may be associated with endogenous OT in the context of intergroup conflict that involves physical harm. We tested this hypothesis by conducting whole-brain analyses of neural responses to perceived ingroup pain. Specifically, we searched for brain regions in which salivary levels of OT predicted activity in response to ingroup pain caused by a member of the outgroup. Our whole-brain analyses revealed that salivary levels of OT were associated with mPFC activity in the Revenge group. Accordingly, we further tested whether the mPFC activity predicted propensity to punish outgroup members and mediated the association between endogenous OT and revenge propensity. The results of these analyses allowed us to test the association between endogenous OT and mPFC responses to ingroup pain as a neurobiological correlate of revenge propensity during intergroup conflict. To provide a broad test of the neurobiological underpinnings of revenge propensity, we also examined whether the tendency to retaliate against outgroup members who are not directly involved in the conflict, which has been termed 'vicarious retribution' (*Lickel et al., 2006*; *Gelfand et al., 2012*; *Lee et al., 2013*), has the same neurobiological association.

In the text below, after describing our experimental design, we first present behavioral results that show comparable ingroup biases in emotions and attitudes in the Revenge and Control groups. We then examine whether the Revenge group, when compared to the Control group, showed higher endogenous OT levels after witnessing ingroup members' pain caused by outgroup members. Thereafter, we report the results of whole-brain analyses that identified neural responses to outgroup-inflicted ingroup pain, which were predicted by salivary levels of OT. Finally, we report evidence for the association between revenge propensity and mPFC activity in response to ingroup pain caused by the outgroup, as well as evidence that the mPFC activity mediates the association between endogenous OT and revenge propensity. These results together suggest an association between endogenous OT and mPFC activity in response to ingroup pain as a neurobiological correlate of revenge propensity during intergroup conflict.

## Results

### Behavioral paradigm

We developed a new behavioral paradigm to examine neurobiological associations of the key component of revenge behavior (i.e., the punishment of the outgroup in response to physical harm to the ingroup caused by the outgroup) while the effects of ingroup biases in emotions and attitudes were controlled. We adopted a between-subjects design by recruiting two independent samples of healthy adults to test our hypotheses.

The paradigm for the Revenge group (n = 40, all males) had three phases and six players (four participants and two confederates). In *Phase 1*, the six players played a game to form an ingroup and an outgroup. Each group consisted of one confederate and two participants (see *Figure 1A*). *Phase two* introduced initial conflict by inviting the participants to watch an ingroup member (Involved_Ingroup target) and an outgroup member (Involved_Outgroup target), both played by the confederates, interacting in a competitive game, during which the winner gave painful or non-painful electric shocks to the rival. In *Phase 3*, the participants underwent fMRI scanning. In four scans, they were informed that Involved_Ingroup and Involved_Outgroup targets continued the competitive game and applied shocks to each other. During each trial, the participants first viewed a photo of the Involved_Ingroup or the Involved_Outgroup target to indicate that the loser of one trial and had to judge his group identity (i.e., ingroup or outgroup) by pressing a button. A lightning (or round)

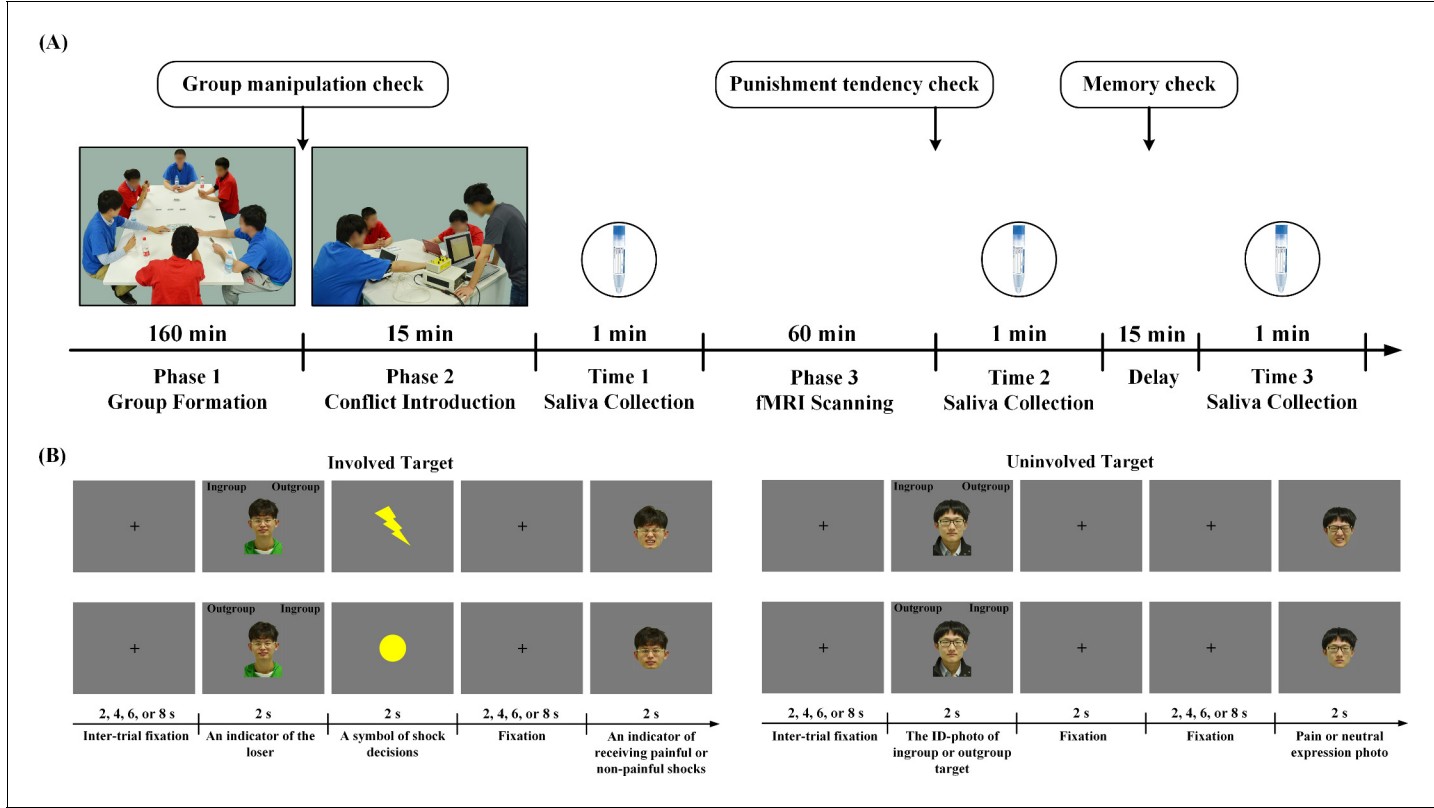

**Figure 1.** Experimental procedure and behavioral results. (**A**) Experimental procedure. Phase 1 assigned four participants and two confederates into two groups who played a game to create group affiliation. In Phase two, the experimenter (in a grey T-shirt) introduced a participant (in a red T-shirt facing the experimenter) to witness a conflict between an ingroup member and an outgroup member (both played by the confederates) who played a competitive game. During fMRI scanning (Phase 3), the participant witnessed ingroup and outgroup members who were directly involved or uninvolved in conflict. Saliva was collected at three points in time. (**B**) Trial structure during fMRI scanning. An ID-photo of Involved_Ingroup or Involved_Outgroup target indicated the loser of the game and the participant had to judge his group identity. A yellow circle or lightning symbol then indicated a non-painful or painful shock. After a fixation, a photo of the loser's face with a painful or neutral expression was displayed to indicate that he was experiencing a painful or non-painful shock. When ID-photos of uninvolved targets were presented, the participant also judged their group identities and passively viewed a following photo of the target with neutral or painful expression.

symbol was then displayed to inform the participant of the winner's decision to give a painful (or non-painful) shock, and this was followed by an image of the target's face with a painful or neutral expression to indicate that the target was experiencing a painful or non-painful shock (*Figure 1B*). Because group members during intergroup conflict are often regarded as an entity of interchangeable members (*Lickel et al., 2006*; *Lee et al., 2013*), we also examined generic neurobiological associations of tendencies to retaliate against outgroup members regardless of their direct involvement in conflict. To this end, in an additional four fMRI scans, the participants were presented with photos of an ingroup and an outgroup member who were not directly involved in the conflict (Uninvolved_Ingroup and Uninvolved_Outgroup targets) and judged their group identity before viewing their painful or neutral expressions. After fMRI scanning, the participants were asked to report how willing they were to punish a target by giving painful shocks (1 = not painful at all, 9 = extremely painful) to estimate their tendencies to punish outgroup members.

We recruited a Control group (n = 40, all males) to control for the effects of perceived group identity, ingroup biases in emotions and attitudes, and ingroup favoritism in empathic brain activity on the potential neurobiological association of revenge propensity. The scenario for the Control group was the same as that for the Revenge group except that, during Phases 2 and 3, the participants were informed that Involved_Ingroup and Involved_Outgroup targets, respectively, played a competitive game with a computer and received painful or non-painful electric shocks given by the computer. Thus, an outgroup bias in tendency to apply painful electric shocks was driven by outgroup derogation in the Control group but by revenge in return for ingroup members' suffering produced by the outgroup in the Revenge group.

We collected saliva from both Revenge and Control groups at three points in time (i.e., Time 1, after introduction of initial intergroup conflict; Time 2, outside the scanner immediately after fMRI scanning; Time 3, 15 min later; *Figure 1A*) to estimate changes of endogenous OT. By comparing the OT results for the Revenge and Control groups, we sought to determine whether endogenous OT increases immediately after initially witnessing an intergroup conflict (Time 1), and whether such effects, if observed, would be enlarged after additional experiences of intergroup conflict (Time 2). We also assessed whether the level of endogenous OT predicted brain responses to perceived ingroup suffering during intergroup conflict. Finally, we examined whether brain responses to ingroup pain mediate the association between endogenous OT and individuals' inclinations to seek revenge by giving painful electric shocks to outgroup members.

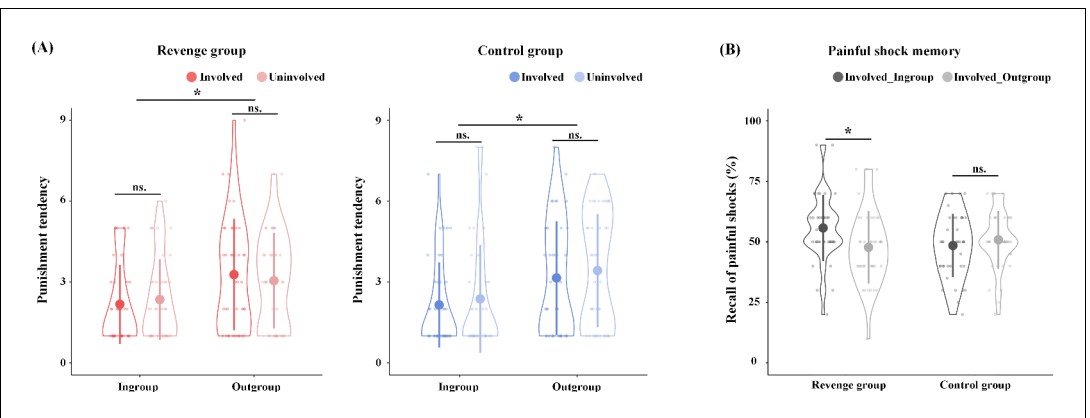

**Figure 2.** Behavioral results. (A) Self-report of punishment tendencies. (B) Memory of painful shocks applied to Involved_Ingroup and Involved_Outgroup targets during scanning. Shown are group means (big dots), standard deviations (bars), measures for each individual (small dots), and distribution (violin shape).
The online version of this article includes the following source data and figure supplement(s) for figure 2:

**Source data 1.** Source data for *Figure 2A*.
**Source data 2.** Source data for *Figure 2B*.
**Figure supplement 1.** Illustration of behavioral results.

## Punishment tendencies in Revenge and Control groups

Revenge and Control groups were matched in age, education, and psychological traits (see *Supplementary file 1* for the demographic information and psychological traits of the participants). In order to assess the effect of the key component of revenge (i.e., the punishment of outgroup members for harm that they inflict on ingroup members), we controlled other concomitant but non-essential factors by collecting self-reports of emotions, attitudes, and punishment tendencies. After the group formation in Phase 1, participants in the Revenge and Control groups reported similar ingroup favoritism in feelings of closeness (Ingroup vs. Outgroup: $4.72 \pm 0.98$ vs. $3.27 \pm 0.99$; $F(1,86) = 136.23$, $p<0.001$, $\eta^2_p = 0.613$). After fMRI scanning, participants were asked to report their emotions and attitudes related to ingroup and outgroup targets on a Likert Scale (1 = not at all, 9 = extremely strong). Participants from both the Revenge and Control groups reported similar ingroup favoritism in emotions and attitudes (see *Figure 2—figure supplement 1*, *Supplementary file 2*, *Supplementary file 3* for statistical details). When viewing ingroup vs. outgroup targets' pain, the participants in both conditions reported greater empathy ($6.58 \pm 1.80$ vs. $6.15 \pm 1.91$; $F(1,78) = 8.21$, $p=0.006$, $\eta^2_p = 0.095$), unpleasantness ($4.23 \pm 2.09$ vs. $3.77 \pm 1.73$; $F(1,78) = 5.37$, $p=0.026$, $\eta^2_p = 0.064$), anger ($2.66 \pm 1.88$ vs. $2.11 \pm 1.35$; $F(1,78) = 9.68$, $p=0.004$, $\eta^2_p = 0.110$), and fear ($2.64 \pm 1.97$ vs. $2.36 \pm 1.72$; $F(1,78) = 3.95$, $p=0.050$, $\eta^2_p = 0.048$, all FDR corrected). By contrast, the participants in both conditions reported greater schadenfreude when viewing outgroup vs. ingroup targets' pain ($2.22 \pm 1.60$ vs. $1.61 \pm 1.00$; $F(1,78) = 14.91$, $p<0.002$, FDR corrected, $\eta^2_p = 0.160$). Moreover, the participants in both conditions reported greater trust ($5.53 \pm 1.38$ vs. $4.51 \pm 1.49$; $F(1,78) = 28.62$, $p<0.002$, FDR corrected, $\eta^2_p = 0.268$) and likability (ingroup: $5.32 \pm 1.53$; outgroup: $4.51 \pm 1.47$; $F(1,78) = 23.57$, $p<0.002$, FDR corrected, $\eta^2_p = 0.232$) for ingroup than outgroup members. Finally, participants in both conditions reported greater tendencies to punish outgroup targets ($3.23 \pm 1.91$ vs. $2.26 \pm 1.53$; $F(1,78) = 27.19$, $p<0.002$, FDR corrected, $\eta^2_p = 0.259$, *Figure 2A*). Importantly, ingroup favoritism in attitudes, emotions, and punishment tendencies did not differ significantly between the Revenge and Control groups and between the involved and uninvolved targets (see *Supplementary file 2* for statistical details). Accordingly, the results of self-report measures indicate that the ingroup/outgroup manipulation was successful in both the Revenge and Control groups. In addition, the results of similar ingroup biases in attitudes, emotions, and punishment tendencies in the two groups suggest that any differences in the neurobiological measures across the Revenge and Control groups cannot be attributed to differences in group affiliation or ingroup favoritism in emotions and attitudes between the two groups.

Previous research has shown that people tend to view their ingroup members as victims and outgroup members as perpetrators during intergroup conflicts (*Ross and Ward, 1995*; *Lickel et al., 2006*). Accordingly, we conducted another manipulation check to assess whether the participants in the Revenge group tended to remember their ingroup members as being the victim of painful shocks during the competitive game to a greater degree than participants in the Control group. After fMRI scanning, participants were asked to recall how often Involved_Ingroup and Involved_Outgroup targets received painful shocks after losing the game. The analysis of variance (ANOVA) of self-report of frequencies of perceived painful shocks, with Intergroup Relationship (ingroup vs. outgroup) as a within-subjects variable and Group (Revenge vs. Control group) as a between-subjects variable, revealed a significant main effect of Intergroup Relationship ($F(1,78) = 5.65$, $p=0.020$, $\eta^2_p = 0.068$). Participants reported a greater number of painful shocks received by ingroup than by outgroup members, even though Involved_Ingroup and Involved_Outgroup targets actually received the same numbers of painful shocks. There was also a significant interaction of Intergroup Relationship x Group ($F(1,78) = 19.21$, $p<0.001$, $\eta^2_p = 0.198$). Simple effect analyses revealed that the Control group reported similar levels of painful shocks that were delivered to Involved_Ingroup and Involved_Outgroup targets ($0.49 \pm 0.13$ vs. $0.51 \pm 0.12$; $t(39) = -1.58$, $p=0.123$, Cohen's $d = 0.25$), whereas the Revenge group reported significantly more painful shocks that were delivered to Involved_Ingroup than to Involved_Outgroup targets ($0.56 \pm 0.14$ vs. $0.48 \pm 0.15$; $t(39) = 4.39$, $p<0.001$; Cohen's $d = 0.69$, *Figure 2B*). The results are consistent with previous findings (*Ross and Ward, 1995*; *Lickel et al., 2006*) and suggest that our manipulations motivated participants of the Revenge (vs. Control) group to view the involved ingroup member more frequently as a victim during intergroup conflict.

## Increased endogenous OT in Revenge groups when compared to Control groups

To test the hypothesis that endogenous OT in humans is sensitive to intergroup conflict, we collected saliva from each participant at three points in time (see *Figure 1A*). If the oxytocinergic system is activated during intergroup conflict in humans, as it is in chimpanzees (*Samuni et al., 2017*), the Revenge (vs. Control) group should show a greater level of OT in saliva at Time one after the initial witness of intergroup conflict. Furthermore, this effect may increase even more at Time two after the participants had witnessed the whole procedure of conflict. We conducted an ANOVA of salivary OT levels with Group (Revenge vs. Control) as a between-subjects variable, and Time (Time-1, -2, and -3) as a within-subjects variable. Because previous research has shown evidence for associations between the administration of OT and ingroup biases in emotions and attitudes (*De Dreu et al., 2011*; *Sheng et al., 2013*), the ANOVA included ingroup biases in feelings of closeness and other emotions and attitudes as covariates. The results showed a significant effect of Group ($F(1,67) =$ 22.66, $p<0.001$, $\eta^2_p = 0.253$) and a significant interaction of Group $\times$ Timing ($F(2,134) = 4.04$, $p=0.020$, $\eta^2_p = 0.057$; *Figure 3A*; see *Table 1* for results of simple effect analyses). These results indicate two important consequences of group conflict: the Revenge (vs. Control) group showed higher endogenous OT levels *immediately after* the initial conflict was observed, and OT levels continued to rise in response to later intergroup conflict in the revenge condition.

To further illustrate the greater increase of OT levels after additional experiences of witnessing intergroup interactions in the Revenge (vs. Control) group, we adopted a standard bootstrapping procedure (*Davison and Hinkley, 1997*) to examine further the difference in increased OT levels between Revenge and Control groups. Specifically, we conducted a bootstrapping analysis to illustrate a greater increase in OT level from Time-1 to Time-2 and a greater decrease of OT level from Time-2 to Time-3 in the Revenge compared to Control group. To do this, we calculated increased OT levels by subtracting measures at Time-1 and Time-3 from those at Time-2 for each participant. Thereafter, a bootstrapped data set in each group was nonparametrically resampled with replacement (i.e., a participant could be selected more than once). The mean of this bootstrapped sample was then calculated and plotted as one of the points (x, y) in a panel with the horizontal (x) and vertical (y) axes showing OT increases measured at Time-2 relative to those at Time-1 and Time-3,

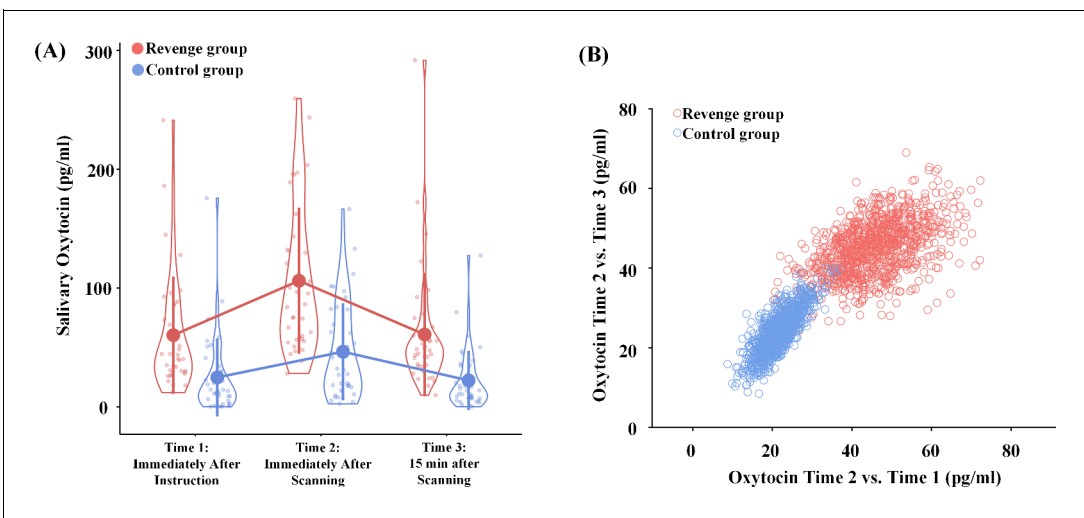

**Figure 3.** Effects of intergroup conflict on endogenous OT. (**A**) Salivary OT levels at three time points during the experimental procedure. Shown are group means (big dots), SD (bars), measures of each individual (small dots), and distribution (violin shape). (**B**) Results of bootstrapping analyses. Increased OT levels were calculated by subtracting either Time-1 and Time-3 measures from the Time-2 measure for the two bootstrapped samples. The online version of this article includes the following source data and figure supplement(s) for figure 3:

**Source data 1.** Source data for *Figure 3A*.
**Source data 2.** Source data for *Figure 3B*.
**Figure supplement 1.** Results of the modified bootstrapping analysis.

**Table 1.** Salivary OT levels (pg/ml) across three time points and the results of simple effect analyses.

| | Revenge (mean ± SD) | Control (mean ± SD) | F | p | $\eta^2_p$ |
|---|---|---|---|---|---|
| OT at Time-1 | 60.40 ± 49.25 | 24.68 ± 32.46 | 10.54 | 0.002 | 0.136 |
| OT at Time-2 | 106.19 ± 61.52 | 47.42 ± 40.83 | 22.90 | <0.001 | 0.255 |
| OT at Time-3 | 60.96 ± 51.92 | 21.72 ± 24.87 | 16.47 | <0.001 | 0.197 |

respectively. The same procedure was repeated 1000 times for each participating group to estimate the population means and variations. As shown in *Figure 3B*, the bootstrapped sample mean points from the Revenge group fall mostly to the upper right of the 2D plot. To confirm the separation of the two bootstrapped samples, we calculated the Euclidean distance between the two samples. The mean distance between the two samples is 32.29, with a 95% confidence interval of 11.55–54.61. We conducted another modified bootstrapping analysis to assess whether the increased OT level related to experiences of conflict that were measured in half of the participants randomly selected from each group can be replicated in the unselected participants. The results suggest similar group differences in increased OT levels that are related to witnessing conflict in the selected and unselected samples (see *Figure 3—figure supplement 1*).

Together, these results support our hypothesis that endogenous OT in humans increases during intergroup conflict. Specifically, the level of endogenous OT seemed to begin to rise immediately after participants initially witnessed intergroup conflict (i.e., at Time-1). Moreover, the OT level increased further after fMRI scanning (i.e., at Time-2) during which the participants had more experiences of intergroup conflict. These findings are consistent with the observations in chimpanzees (*Samuni et al., 2017*), and provide empirical evidence that intergroup conflict in primates including humans is associated with increased levels of endogenous OT.

## Brain responses to perceived pain in the Revenge and Control groups

In our design, an increase in brain activity in response to perceived painful (vs. neutral) expressions is a precondition for examining the revenge-related functional role of the association between endogenous OT and neural responses in either the empathy or theory-of-mind networks. In addition, on the basis of previous findings of ingroup favoritism in empathic neural responses (*Xu et al., 2009*; *Hein et al., 2010*; *Cikara et al., 2011*; *Sheng and Han, 2012*; *Han, 2018*), we expected greater neural responses to ingroup than outgroup pain if our group manipulations were successful. The presence of ingroup favoritism in empathic neural responses provides a precondition for further analyses of OT associations with empathic neural responses to ingroup and outgroup pain separately. Therefore, we first examined participants' neural responses to perceived pain in others by conducting a whole-brain analysis that collapsed all targets and all participants. Similar to previous findings (*Fan et al., 2011*; *Lamm et al., 2011*; *Shamay-Tsoory, 2011*), whole-brain analyses of the contrast of painful vs. neutral expressions revealed activations in both the empathy network, including the anterior cingulate and bilateral AI/inferior frontal gyrus (IFG), and the theory-of-mind network, including the mPFC, left TPJ, and right temporal pole (TP) (all activations were identified by combining a voxel-level threshold of $p<0.001$ and a cluster-level threshold of $p<0.05$, FWE corrected, *Figure 4A*, *Supplementary file 4*, see *Figure 4—figure supplement 1*, *Figure 4—figure supplement 2*, *Supplementary file 5*, *Supplementary file 6* for the results from the Revenge and Control groups, separately).

Separate whole-brain analyses that collapsed all participants in the Revenge and Control groups identified activations in the mPFC, aMCC, bilateral AI/IFG, and left TPJ in response to ingroup targets' pain but only in the mPFC in response to outgroup targets' pain (combined a voxel level threshold $p<0.001$ and a cluster level threshold $p<0.05$, FWE corrected, *Figure 4B*, *Supplementary file 4*). To further examine the ingroup favoritism in neural responses in these brain regions, we conducted region-of-interest (ROI) analyses of neural responses to others' pain in the brain regions identified in the whole-brain analyses. ROI were defined as spheres with 5 mm radius centered at the peak activation using a leave-one-out method by collapsing all participants. The leave-one-out method identified activations in the bilateral AI/IFG, mPFC, and left TPJ in response to painful vs. neutral expressions at the combined voxel level threshold $p<0.001$ and cluster level

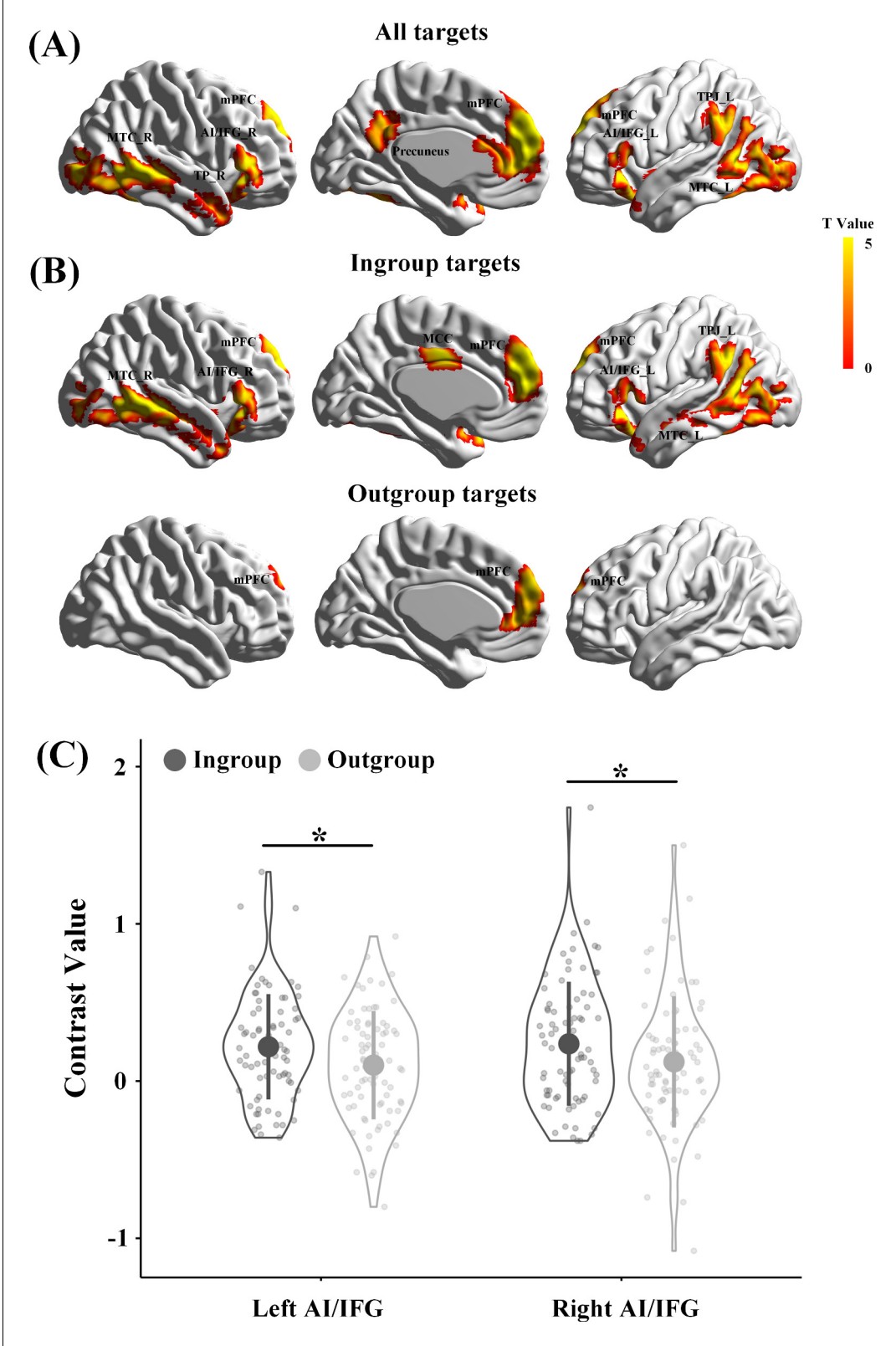

**Figure 4.** Brain activities in response to painful vs neutral expressions. (A) Illustrations of brain responses to painful vs. neutral expressions of all targets perceived during scanning across all participants from Revenge and Control groups and collapsing involved and uninvolved targets. (B) Illustrations of brain responses to painful vs. neutral expressions of ingroup targets and outgroup targets across all participants. (C) The contrast values of neural

*Figure 4 continued on next page*

*Figure 4 continued*

responses to painful (vs. neutral) expressions in the left and right AI/IFG. Shown are group means (big dots), SD (bars), measures of each individual (small dots), and distribution (violin shape). mPFC, medial prefrontal cortex; MCC, midcingulate cortex; MTC, middle temporal cortex; AI/IFG, anterior insula and inferior frontal cortex; TPJ, temporoparietal junction; TP, temporal pole.

The online version of this article includes the following source data and figure supplement(s) for figure 4:

**Source data 1.** Source data for *Figure 4A*.
**Source data 2.** Source data for *Figure 4B* _Ingroup.
**Source data 3.** Source data for *Figure 4B* _Outgroup.
**Source data 4.** Source data for *Figure 4C*.
**Figure supplement 1.** Illustrations of brain responses to painful vs. neutral expressions across participants and collapsed involved and uninvolved targets in the Revenge group (a voxel level threshold p<0.001, uncorrected and a cluster level threshold of p<0.05, FWE corrected).
**Figure supplement 2.** Illustration of brain responses to painful vs. neutral expressions across participants and collapsed involved and uninvolved targets in the Control group (a voxel level threshold p<0.001, uncorrected and a cluster level threshold of p<0.05, FWE corrected).

---

threshold $p<0.05$, FWE corrected. The contrast values (painful vs. neutral expressions) were extracted from each ROI and subjected to ANOVAs with Relationship (Ingroup vs. Outgroup) and Involvement (Involved vs. Uninvolved) as within-subjects variables and Group (Revenge vs. Control group) as a between-subjects variable. The results confirmed greater neural responses to ingroup than outgroup targets' pain in the empathy network, including the left IFG/AI ($0.22 \pm 0.34$ vs. $0.10 \pm 0.34$, $F(1,78) = 5.41$, $p=0.036$, $\eta^2_p = 0.065$, all results of ROI analyses were FDR corrected) and right IFG/AI ($0.24 \pm 0.39$ vs. $0.12 \pm 0.42$, $F(1,78) = 4.56$, $p=0.036$, $\eta^2_p = 0.055$, *Figure 4C*), but not in the theory-of-mind network (mPFC, $0.33 \pm 0.45$ vs. $0.20 \pm 0.44$, $F(1,78) = 3.92$, $p=0.102$, $\eta^2_p = 0.048$; left TPJ, $0.21 \pm 0.46$ vs. $0.13 \pm 0.43$, $F(1,78) = 1.45$, $p=0.232$, $\eta^2_p = 0.018$). The effect of increased neural responses to ingroup targets' pain (vs outgroup targets' pain) did not differ significantly between Revenge and Control groups (left IFG/AI, $F(1, 78)=0.09$, $p=0.770$, $\eta^2_p = 0.001$; right IFG/AI, $F(1, 78)=0.13$, $p=0.716$, $\eta^2_p = 0.002$) and between Involved and Uninvolved targets (left IFG/AI, $F(1,78) = 2.71$, $p=0.104$, $\eta^2_p = 0.034$; right IFG/AI, $F(1,78) = 2.68$, $p=0.105$, $\eta^2_p = 0.033$).

These results replicate the previous neuroimaging findings of activations in the empathy and theory-of-mind networks in response to perceived pain in others (*Fan et al., 2011*; *Lamm et al., 2011*; *Shamay-Tsoory, 2011*), and of enhanced neural responses to ingroup rather than outgroup pain (*Xu et al., 2009*; *Hein et al., 2010*; *Cikara et al., 2011*; *Sheng and Han, 2012*; *Han, 2018*). These results provide bases for further tests of the association between endogenous OT and brain responses to perceived pain in others. Importantly, the results provide no evidence for difference in ingroup favoritism in empathic neural responses between the Revenge and Control groups. Therefore, any possible contribution of ingroup biases in brain activity to group differences in endogenous OT and associations between endogenous OT and brain responses to others' pain was reduced to a minimum degree.

## Endogenous OT predicts mPFC activity in response to ingroup pain

If the association between endogenous OT and brain responses to ingroup pain serves as a neurobiological correlate of revenge propensity during intergroup conflict, endogenous OT after the initial intergroup conflict at Time-1 should predict subsequent brain responses to ingroup pain, which may then further predict revenge propensity. Accordingly, we first conducted a whole-brain regression analysis to examine whether OT levels at Time-1 predicts the brain responses to perceived ingroup pain. As discussed below, this analysis revealed an association between the mPFC activity and OT level at Time-1 in the Revenge group. In order to then estimate whether the OT-mPFC association was specific to OT levels at Time-1, we conducted a second whole-brain regression analysis to examine brain responses to ingroup pain that were associated with endogenous OT measured after fMRI scanning at Time-2. Whole-brain analyses were used so as to not bias OT association with a specific network (e.g., the empathy or theory-of-mind network).

In the first whole-brain regression analysis, the OT level at Time-1 was entered into a general linear model as predictors of brain responses to painful (vs. neutral) expressions of each target. The

results showed that, for the Revenge (but not the Control) group, OT level at Time-1 reliably predicted the mPFC activity in response to Involved_Ingroup target's pain (combined a voxel level threshold p<0.001 and a cluster level threshold p<0.05, FWE corrected, *Figure 5A*). We conducted ROI-based moderation analyses to further confirm the group differences in the coupling between the OT level at Time-1 and mPFC activity. The mPFC activity to Involved_Ingroup targets' painful (vs. neutral) expression was extracted from the ROI defined in the leave-one-out whole-brain analysis of the contrast of painful vs. neutral expressions using a threshold that combined a voxel level threshold p<0.001 and a cluster level threshold p<0.05, FWE corrected. The mPFC activity to Involved_Ingroup targets' pain was entered as the independent variable, Group (Revenge vs. Control) was entered as the moderator, and ingroup biases in closeness, emotion and attitudes were entered as covariates that had possible contributions to the association between OT levels and mPFC activity during intergroup conflict. The moderation analysis including the covariates showed that the interaction between mPFC activity and Group accounted for a significant proportion of variance in the OT level at Time-1 ($\Delta R^2$=0.05, $\Delta F(1,65)$=5.70, p=0.03; *Figure 5B and C*, see *Supplementary file 7* for statistical details). The results suggest that the association between endogenous OT and mPFC activity was specific to Revenge group.

Furthermore, to test whether the direct involvement of an ingroup member in the conflict was critical for the association between endogenous OT and mPFC activity in response to ingroup pain,

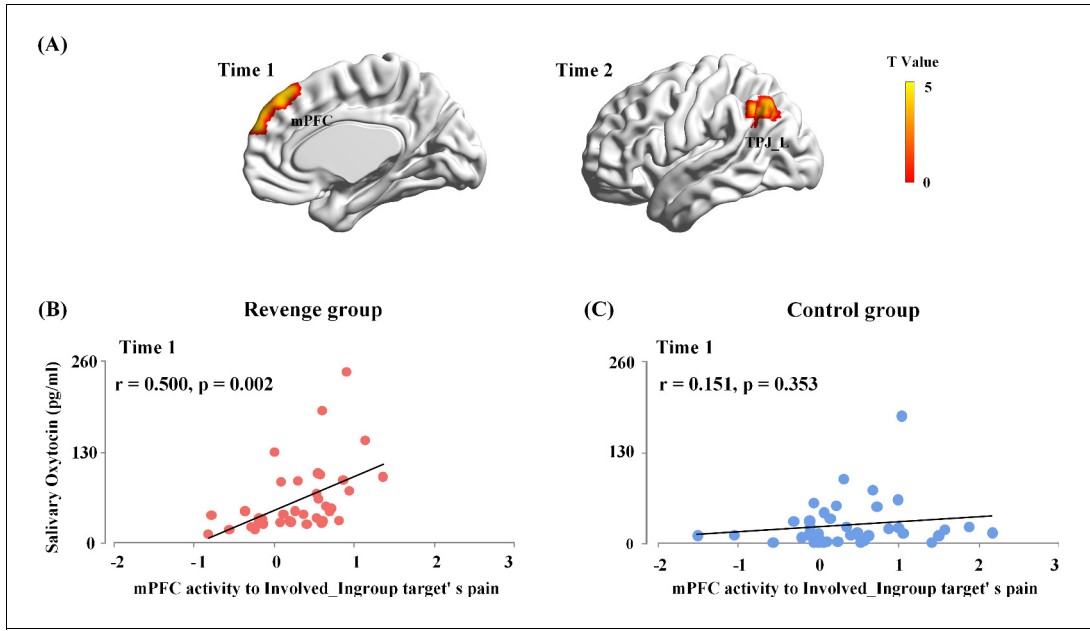

**Figure 5.** Associations between endogenous OT and brain activity in response to others' suffering. (**A**) The mPFC activity to Involved_Ingroup target's pain associated with the endogenous OT at Time-1 in the Revenge group. The OT level at Time-2 reliably predicted the left TPJ activities in response to the Involved_Ingroup target's pain in the Revenge group. A voxel-level threshold of p<0.001 and a cluster-level threshold of p<0.05, FWE corrected, was used to identify and to visualize brain activations. (**B**) The associations between endogenous OT-levels at Time-1 with the mPFC activity in response to the Involved_Ingroup target's pain for the Revenge group. (**C**) No significant correlation between endogenous OT-levels and the mPFC activity in response to Involved_Ingroup target's pain was found for the Control group. Note: the results of the moderation analysis indicate a significant group difference in the association between endogenous OT-levels at Time-1 and the mPFC activity in response to Involved_Ingroup target's pain.

The online version of this article includes the following source data for figure 5:

**Source data 1.** Source data for *Figure 5A* _Time 1.
**Source data 2.** Source data for *Figure 5A* _Time 2.
**Source data 3.** Source data for *Figure 5B*.
**Source data 4.** Source data for *Figure 5C*.

we conducted an additional ROI-based moderation analysis to examine the differential coupling between OT level at Time-1 and mPFC activity in response to Involved_Ingroup vs. Uninvolved_Ingroup targets' pain in the Revenge group. The moderation analysis used the repeated measures of mPFC activities towards Involved_Ingroup vs. Uninvolved_Ingroup targets as the moderator. The results showed that the endogenous OT level at Time-1 accounts for a significant amount of variance in mPFC activities in response to Involved_Ingroup vs. Uninvolved_Ingroup targets' pain ($R^2$=0.15, F (1, 75)=12.99, p<0.001). The results suggest a stronger coupling between endogenous OT and mPFC activity in response to Involved_Ingroup (compared to Uninvolved_Ingroup) targets' pain during intergroup conflict.

In the second whole-brain regression analysis, OT level at Time-2 was entered into a general linear model as a predictor of brain responses to the painful (vs. neutral) expressions of each target. The results showed that OT level at Time-2 was significantly associated only with the left TPJ activity in response to Involved_Ingroup target's pain (combining a voxel level threshold p<0.001 and a cluster level threshold p<0.05, FWE corrected, *Figure 5A*). However, ROI-based moderation analyses failed to confirm any significant differences between the responses of the Revenge and Control groups in terms of the coupling between the OT level at Time-2 and left TPJ activity. Thus, the results provide no evidence to support a revenge-specific association between brain responses to ingroup pain and further changes in endogenous OT.

Together, these results suggest that intergroup conflict enhanced the link between endogenous OT measured after the initial experience of ingroup conflict and mPFC activity in response to the perceived pain of the ingroup member who was directly involved in conflict with an outgroup member. These results provide bases for further examination of the functional role of the mPFC activity in mediating the association between endogenous OT after initial experience and revenge propensity.

## Association between mPFC activity and revenge propensity

Because only the mPFC activity in response to ingroup members' pain was coupled with endogenous OT level at Time-1, we conducted an ROI analysis to examine the associations between the mPFC activity to ingroup members' pain and tendencies of both the Revenge and Control groups to punish outgroup members. The contrast values of painful vs. neutral expressions of Involved_Ingroup targets were extracted from an ROI (a sphere with 5 mm radius) centered at the mPFC activation (using a leave-one-out method by collapsing participants from the two subject groups). The results of correlation analyses showed that, for the Revenge (but not the Control) group, the mPFC activity in response to Involved_Ingroup targets' pain positively predicted punishment tendencies toward both Involved_Outgroup and Uninvolved_Outgroup targets (r = 0.35 and 0.42; p=0.026 and 0.014, FDR corrected, *Figure 6A*). The results suggest that individuals with stronger mPFC activity in response to ingroup pain tended to apply more painful shocks to outgroup members, regardless of whether they were directly involved in the conflict. This finding provides a potential neural basis for understanding how conflicts between two individuals spread across the two groups with which the two individuals are affiliated (*Gelfand et al., 2012*; *Lee et al., 2013*). In addition, the finding of the association between mPFC activity and revenge propensity provides a basis for the following mediation analysis.

Finally, we estimated the neurobiological (from endogenous OT to mPFC activity in response to ingroup pain) association of revenge propensity during intergroup conflict by conducting ROI-based mediation analyses in the Revenge group. The analyses focused on the functional role of mPFC activity to ingroup pain in mediating the relationship between endogenous OT measured after the initial intergroup conflict (Time-1) and later punishment tendencies toward outgroup. The first mediation analysis examined whether the mPFC activity in response to Involved_Ingroup targets' pain mediates the relationship between the OT level at Time-1 and the tendency to punish Involved_Outgroup targets. In Step 1 of the mediation model, the regression of the OT level on punishment tendency toward an Involved_Outgroup target was not significant (b = 0.16, t(35) = 0.94, p=0.355) when not considering the mediator (e.g., the mPFC activity). Step two showed that the regression of the OT level on the mediator was significant (b = 0.50, t(35) = 3.41, p=0.002). Step three showed that the regression of the mediator on punishment tendency was significant (b = 0.38, t(34) = 2.05, p=0.048) when controlling for the OT level. Step four revealed that the OT level was not a significant predictor of punishment tendency (b = −0.03, t(34) = −0.17, p=0.863) when controlling for the mediator

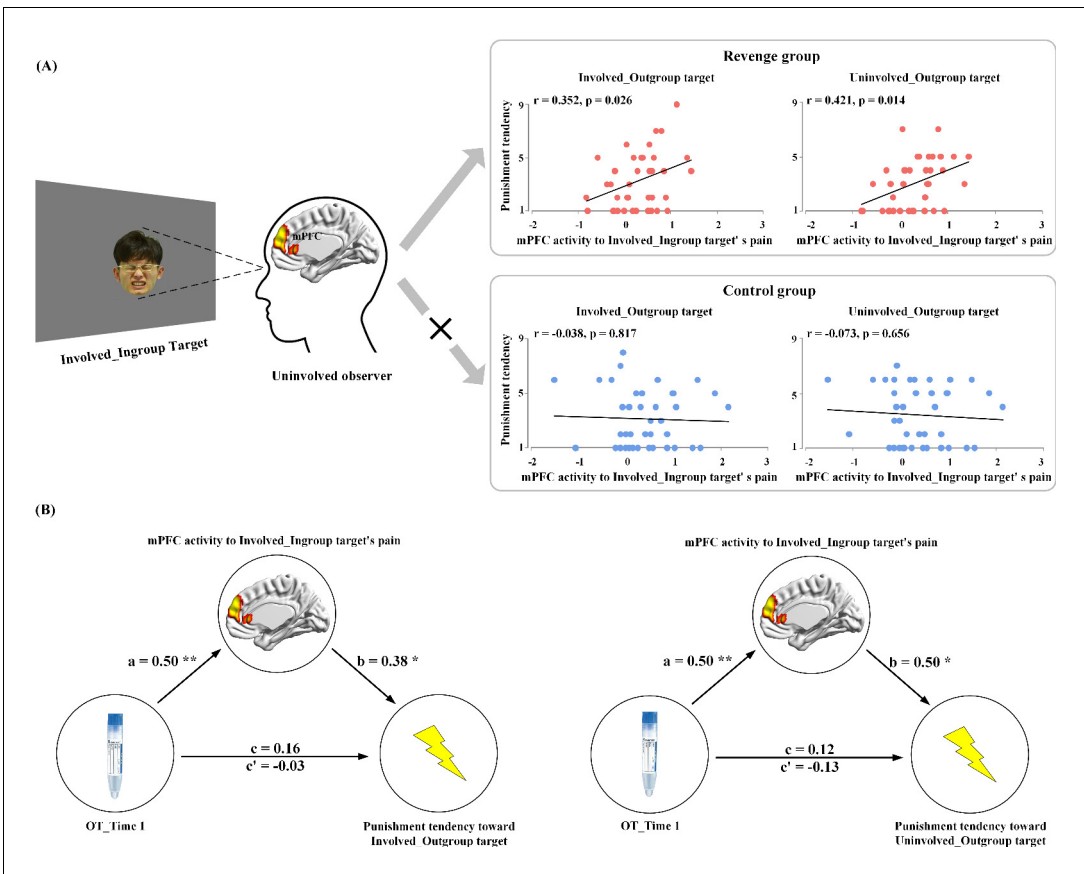

**Figure 6.** Results of brain-propensity associations and mediation analyses. (**A**) Brain-propensity associations in the Revenge group. The mPFC activity in response to Involved_Ingroup target's pain in an uninvolved observer from the Revenge (but not the Control) group predicted his punishment tendencies toward both Involved_Outgroup and Uninvolved_Outgroup targets. (**B**) The mPFC mediation of endogenous OT and punishment tendencies. The mPFC activity to Involved_Ingroup targets' pain mediates the relationship between the salivary level of endogenous OT at Time-1 and punishment tendencies toward Involved_Outgroup targets (left) and Uninvolved_Outgroup targets (right). mPFC mediation of association between endogenous OT and outgroup punishment.

The online version of this article includes the following source data for figure 6:

**Source data 1.** Source data for *Figure 6A*.

---

(*Figure 6B*, see *Supplementary file 8* for statistical details). The indirect effect size was 0.19 with a 95% confidence interval that did not include zero (0.01–0.44).

The second mediation analysis examined whether the mPFC activity in response to Involved_Ingroup targets' pain mediates the relationship between the OT level at Time-1 and tendency to punish Uninvolved_Outgroup targets. In step 1 of the mediation model, the regression of the OT level on punishment tendency toward the Uninvolved_Outgroup target was not significant (b = 0.12, t(35) = 0.70, p=0.488) when not considering the mediator (e.g., the mPFC activity). Step two showed that the regression of the OT level on the mediator was significant (b = 0.50, t(35) = 3.41, p=0.002). Step three showed that the regression of the mediator on punishment tendency was significant (b = 0.50, t(34) = 2.80, p=0.008) when controlling for the OT level. Step four revealed that the OT level was not a significant predictor of punishment tendency (b = −0.13, t(34) = −0.74, p=0.467) when controlling for the mediator (*Figure 6B*, see *Supplementary file 9* for statistical details). The indirect effect size was 0.25, with a 95% confidence interval that did not include zero (0.08–0.46). These results indicate that the mPFC activity in response to ingroup pain caused by an outgroup mediates the association between endogenous OT measured after initially witnessing intergroup conflict and tendencies to retaliate upon outgroup members, regardless of whether they directly

brought physical harm to ingroup members. The results of these mediation analyses provide additional evidence for the endogenous-OT/mPFC association as a neurobiological correlate of revenge propensity during intergroup conflict.

## Discussion

Revenge behavior is universal and highly costly. Accordingly, understanding its neurobiological bases is of critical theoretical and practical importance. Revenge behavior during intergroup conflict engages multiple psychological processes from perceiving ingroup suffering to making aggressive decisions toward outgroups. Our research focused on how neurobiological responses to ingroup pain are related to revenge decisions during intergroup conflict. Although multiple motives are involved in vengeful behaviors during intergroup conflict, we compared salivary OT and brain activity in the Revenge and Control groups. This design allowed us to isolate the neurobiological responses to perceived ingroup suffering resulting from physical harm caused by an outgroup member and its association with seeking revenge through physical harm upon outgroup members.

Previous research has reported endogenous OT reactivity during intergroup conflict in wild chimpanzees (*Samuni et al., 2017*), but we showed the first evidence of increased levels of endogenous OT in humans during intergroup conflict that involves physical harm to ingroups caused by an outgroup. The salivary OT level in humans is affected by affiliative contact and is interrelated with the plasma OT level (*Feldman et al., 2011*). Our results complement previous research on the effect of intranasal OT administration on ingroup cooperation and outgroup defensive competition in economic games (*De Dreu et al., 2010*; *De Dreu et al., 2011*). Importantly, our findings suggest that endogenous OT is an influential physiological mechanism in humans that is activated in response to intergroup conflicts that involve physical harm between ingroup and outgroup members. It is worth noting that, similar to the finding in chimpanzees (*Samuni et al., 2017*), our results suggest that endogenous OT increases after initially witnessing intergroup conflict (e.g., at Time-1). It appears that, in both humans and chimpanzees, the oxytocinergic system quickly responds to intergroup conflict. In addition, the endogenous OT increased more after further witnessing intergroup conflict (e.g., at Time-2) and dropped when intergroup conflict had ended (e.g., at Time-3). These results illustrate dynamic changes of endogenous OT that occur throughout the entire duration of an intergroup conflict.

Our results also revealed intermediate brain mechanisms linking endogenous OT reactivity to perceived intergroup conflict and revenge propensity during intergroup conflict. Unlike previous research that focused on increased neural activity following aggressive decisions (*Seymour et al., 2007*; *Krämer et al., 2007*; *Krueger and Hoffman, 2016*; *Chester and DeWall, 2016*), our work showed that the mPFC activity in response to ingroup pain predicted the propensity for subsequent revenge behavior during intergroup conflict. The mPFC is well-known for its functional role in representing mental states (*Amodio and Frith, 2006*), social emotion (*Harris and Fiske, 2007*; *Mathur et al., 2010*), and group identity (*Volz et al., 2009*; *Molenberghs and Morrison, 2014*). Our results further revealed that the neurobiological association between endogenous OT and mPFC activity in response to Involved_Ingroup targets' suffering is related to propensity to punish outgroup members, regardless of whether the outgroup members were directly involved in the conflict. This cross-group brain-propensity association is different from previous findings of a within-group brain-propensity association in an intergroup context without direct conflict. That is, neural responses to ingroup pain predict tendencies to help ingroup members, or neural responses to outgroup pain predict tendencies not to help outgroup members (*Hein et al., 2010*; *Cikara et al., 2011*; *Mathur et al., 2010*). Our results cast a new perspective on the neural underpinnings that drive decisions to apply physical harm toward outgroups during intergroup conflict. More generally, the cross-group brain-propensity association suggests a potential neural mechanism underlying the *contagion* of revenge behavior, and may help us to understand why disputes between two individuals can escalate across groups and across time (*Gelfand et al., 2012*).

Our findings make fundamental contributions to the intergroup conflict literature and, in particular, towards understanding the neurobiological associations of ingroup love and outgroup hate. Intergroup conflict plays a substantial role in the evolution of both aggressiveness against outgroup and cooperativeness towards ingroups (*Rusch, 2014*). Although previous studies have demonstrated the role of OT and mPFC activity in altruistic decisions favoring the ingroup (*De Dreu et al., 2010*;

*Mathur et al., 2010*; *De Dreu et al., 2011*), our results suggest that the context of intergroup conflict may shift the key function of the oxytocinergic system from mediating ingroup love (a desire to help the ingroup) to facilitating outgroup hate (an aggressive motivation to hurt the outgroup). Such variation in the social function of OT may assist individuals to adapt to changing social contexts (*Shamay-Tsoory and Abu-Akel, 2016*; *Ma et al., 2016*). Unlike previous research that focused on increased activity in the reward system as a consequence of aggressive decisions (*Krämer et al., 2007*; *Chester and DeWall, 2016*), our findings highlight a neurobiological association between endogenous OT and the mPFC that occurs prior to but is linked to revenge propensity during intergroup conflicts. Our results open a new avenue toward understanding the neurobiological mechanisms that mediate aggression-related hormones and social decisions related to intergroup hostility, and provide a neuroscientific account of revenge motives during intergroup conflict.

The revenge propensity shown in the Revenge group cannot simply be explained by OT-induced negative emotions. Increasing evidence suggests that the oxytocinergic system is involved in modulating multiple social emotions that are either positively (e.g., empathy, *Sheng et al., 2013*) or negatively (e.g., schadenfreude, *Shamay-Tsoory et al., 2009*) related to social behaviors. Our participants from both the Revenge and Control groups reported greater schadenfreude when viewing outgroup suffering rather than ingroup suffering. Schadenfreude has been linked to striatum activation induced by misfortunes happening to envied persons (*Takahashi et al., 2009*). Although previous research has shown that viewing outgroup pain can activate the nucleus accumbens (NAcc) and greater NAcc activity predicted less motivation to help outgroup members (*Hein et al., 2010*; *Luo et al., 2015*), neither the Revenge group nor the Control group in our work showed activations in the reward system when viewing outgroup members' suffering. Thus, negative emotion such as schadenfreude may play a minimal role in modulating revenge propensity in our experimental settings.

Although our measures of endogenous OT and brain activity suggest that the association between endogenous OT and mPFC activity in response to perceived ingroup suffering is related to revenge tendencies, we noted that the latter were estimated by self-reports. It is unclear whether such measures were actually correlated with revengeful behavior. To test this, we conducted an independent behavioral experiment in a new sample (see *Supplementary files 10* and *11*). The experimental procedures were the same as those in our fMRI experiment except that Phase three was modified in the following way. While viewing Involved_Ingroup and Involved_Outgroup targets who played the competitive game with each other (Revenge group) or with a computer (Control group), participants were occasionally (in four trials, two on Involved_Ingroup targets and two on Involved_Outgroup targets) asked to make punishment decisions by deciding the intensity on a Likert Scale (1 = not painful at all, 9 = extremely painful) of electric shocks that were believed to be inflicted on the targets. To enhance participants' beliefs about the experimental setting, they only viewed Involved_Ingroup and Involved_Outgroup targets during the game and they made punishment decisions simultaneously but in different rooms. Moreover, after each punishment decision, a feedback face with either painful or neutral expression, depending on a participant's decision, was presented to inform the participant of the consequence of his decision. The results showed evidence that the measures of punishment tendencies were positively correlated with the measures of actual punishment decisions toward Involved_Outgroup target in both Revenge and Control groups (r = 0.75 and 0.70, FDR corrected ps <0.001). These results indicate that our measures of punishment tendencies can, to a certain degree, reflect individuals' punishment decisions with real consequences.

Our findings also raise new questions about the role of other brain regions in the process of revenge. For example, recent research has shown that, in a scenario in which an observer punishes transgressors due to social norm violation (i.e., third-party punishment), the willingness to punish severely was associated with increased amygdala activity (*Stallen et al., 2018*), possibly reflecting the encoding of affective arousal associated with harm done to someone else (*Buckholtz and Marois, 2012*; *Krueger and Hoffman, 2016*). The current work, however, did not find evidence of an association between amygdala activity and punishment tendencies during intergroup conflict. It is possible that punishment decisions toward outgroup in the context of intergroup conflict are justified as revenge that reduces ingroup suffering and thus bring less negative arousal. Future research is needed to support this speculation.

Our work also expands the literature to examine neural responses that are implicated in vicarious revenge, which occurs when a person punishes an outgroup member who is not one of the direct causal agents in the original attack against an ingroup member (*Lickel et al., 2006*; *Gelfand et al., 2012*; *Lee et al., 2013*). Neither the agent of retaliation nor the target of retribution is directly involved in the original conflict during vicarious retribution, similar to the punishment of Uninvolved-Outgroup targets in our work. We showed that the mPFC activity in response to ingroup pain similarly predicted punishment tendencies toward Involved-Outgroup targets and Uninvolved-Outgroup targets. The mPFC activity also mediated the relationship between endogenous OT and tendency to punish Involved_Outgroup targets, as well as the relationship between endogenous OT and tendency to punish Uninvolved_Outgroup targets. Thus, our findings suggest a neurobiological correlate of punishment tendency during intergroup conflict that does not differentiate between direct and indirect vicarious retribution. This is a possible result of the fact that outgroup members are perceived as a unified and coherent entity and share the same blameworthy qualities during intergroup conflict (*McConnell et al., 1997*; *Crawford et al., 2002*; *Lee et al., 2013*).

In conclusion, by integrating a neural-behavioral paradigm with fMRI, we provided evidence that intergroup conflict is associated with increased salivary levels of OT in humans, which further predicted stronger mPFC activity in response to ingroup suffering caused by an outgroup member. Moreover, the mPFC activity mediates the association between endogenous OT and propensity to seek revenge by giving painful electric shocks to outgroup members. Our findings highlight the coupling of the OT system and the mPFC as a neurobiological correlate of revenge propensity during intergroup conflict. Our paradigm can be applied to other samples (e.g., females) and cultures (e.g., where individualism is more dominant) to advance our understanding of the neurobiological underpinnings of revenge propensity and behavior during intergroup conflict.

Finally, because other motivations also drive revenge behavior in intergroup contexts, including feeling threat to group pride (*Turner and Tajfel, 1986*), empathy for the harmed ingroup members (*Smith et al., 1999*; *Davis, 2018*), and normative pressure to avenge the ingroup (*Deutsch and Gerard, 1955*), future research should examine different motivations driving revenge and concomitant emotions that become activated in a host of revenge situations.

## Materials and methods

### Participants

Our fMRI experiment recruited 44 male Chinese university students for Revenge Group (mean age ± SD = 23.27 ± 2.76 years) and 44 male Chinese university students for Control Group (mean age ± SD = 23.89 ± 2.16 years). Four participants from each group were excluded from fMRI data analyses due to their excessive head movements during scanning, leaving 40 participants in each group being included for data analyses (Revenge Group: mean age ± SD = 23.20 ± 2.78 years; Control Group: mean age ± SD = 23.70 ± 2.03 years). Our behavioral experiment recruited independent samples of 40 male Chinese university students for the Revenge Group (mean age ± SD = 22.50±2.75 years) and 39 male Chinese university students for the Control Group (mean age ± SD = 21.56 ± 2.23 years, one participant from the Control group dropped out and was substituted by an additional confederate). Demographic information and psychological traits of Revenge and Control groups are shown in *Supplementary file 1*. All participants were right-handed, had normal or corrected-to-normal vision and reported no neurological or psychiatric history. All participants were paid for their participation. Informed consent was obtained from all participants prior to the experiment. Experimental protocols were approved by the Research Ethics Committee at the School of Psychological and Cognitive Sciences (#2015-12-04), Peking University, complying with the Declaration of Helsinki. The images used in *Figures 1* and *6* are photographs of the confederates and the consent to publish these images was obtained.

The sample size was estimated using G*Power (*Faul et al., 2009*). Because we aimed to assess the association between brain activity in response to perceived ingroup members' suffering and retaliation upon outgroup members, the first power analysis estimated the sample size that allowed the detection of reliable brain activities in response to others' pain (e.g., the contrast of painful vs. neutral expressions). On the basis of the previous fMRI study of empathy for pain (*Han et al., 2017*), the effect size of brain activities (including aMCC, bilateral AI and bilateral SII) in response to others'

suffering was between 0.39 and 0.84 (a middle effect size). On the basis of G*Power estimation, a sample size of 34 participants for each group was required to obtain a middle effect size of 0.5 with an error probability of 0.05 and power of 0.80 in paired t-tests (two-tails).

Because there is no previous research allowing us to conduct an experience-based estimation of the effect size of Revenge/Control group difference in salivary OT level, we selected a middle effect size of 0.25 for sample size estimation. To test the difference in endogenous OT levels between the Revenge and Control groups, we planned to conduct an ANOVA with Time (Time-1, -2, -3) as a within-subjects variable and Group (Revenge vs. Control) as a between-subjects variable. To detect a significant main effect of Group required a total sample size of 86 with an error probability of 0.05 and power of 0.8, given the correlation among repeated measures (0.5). To detect a significant interaction between Time and Group required a total sample size of 28 with an error probability of 0.05 and power of 0.8, given the correlation among repeated measures (0.5) and the nonsphericity correction (1).

## Behavioral and imaging procedures

On each testing day, four participants and two confederates were recruited. Each band of four participants was alternately assigned to the Revenge or Control group in order to balance the sample size of the two groups. Participants and confederates had not known each other before their participation. The experimental procedure consisted of three phases starting at 9:00 am on the testing day.

### Phase 1: group formation

Upon arrival at a testing room, three photos were taken from each participant (including confederates). One ID photo with neutral expression was used for estimation of attitudes and judgments of group identity. The ID photo and other two photos with neutral or painful expressions were used during fMRI scanning (*Figure 1*). The photos were taken by asking participants to show a neutral expression or a painful expression (asking participants to imagine a painful experience), which was characterized by facial movements including brow lowering, orbit tightening, and raising of the upper lip (*Prkachin, 1992*). The photos from all participants were modified to the same size (400 × 600 pixels for ID-photos and 400 × 500 pixels for the two photos with neutral or painful expressions).

All participants including confederates were asked to complete questionnaires to estimate self-esteem (*Rosenberg, 1965*), extroversion-introversion (*Eysenck and Eysenck, 1975*), self-construal (*Singelis, 1994*), individualism/collectivism (*Triandis and Gelfand, 1998*), trait empathy (*Davis, 1983*), and trait aggression (*Buss and Perry, 1992*). Subjective socioeconomic status was assessed using a ladder with 10 rungs (*Kilpatrick and Cantril, 1960*). The participants were informed that they would be divided into two groups on the basis of the results of questionnaire measures, though they were actually randomly assigned to two groups so that there were one confederate and two participants in each group.

Participants from each group were asked to wear T-shirts of the same color (red or blue, *Figure 1A*). Participants introduced their own names, nicknames, majors, and hobbies so that they became familiar with each other. Participants then started to play the *Saboteur* card game (http://www.annarbor.com/entertainment/saboteur-card-game-review/). During this game, ingroup members played cards to build a tunnel to a destination where gold is located or to block the tunnel to prevent outgroup members from reaching the goal. This game required ingroup members to cooperate with each other but to interfere with outgroup members so as to reach the destination before the outgroup. The intergroup relationship was built by playing this game for 90 min. To check the effectiveness of the group manipulation, after the game, participants were asked to complete a modified version of the Inclusion of Other in the Self Scale (*Aron et al., 1992*) to assess their feelings of closeness between oneself and ingroup members, and between oneself and outgroup members. Phase 1 lasted for 160 min.

### Phase 2: inducing intergroup conflict

After Phase 1, a participant was led to another test room where the two confederates in representation of each group were supposed to be playing a competitive game. During this game, the two

confederates performed the classic Stroop task (*Stroop, 1935*) by responding to colors of words by button presses. Participants from the Revenge Group were informed that the two confederates (one from the ingroup and one from the outgroup) competed with each other to make the most correct responses. After three trials, the winner who made more correct responses or responded faster then decided whether to give the rival a painful electric shock (as an index of aggression). A pair of foil electrodes connected to an instrument (DS7A Digitimer) for generation of electric shocks and the left hand of each confederate. The participants witnessed that the confederate who won first pressed a button on the instrument to give a non-painful shock to the loser and the confederate who lost showed a neutral expression. Another confederate who won later, however, chose to give a painful shock to the loser by saying 'I am curious about how painful an electric shock can be'. The confederate who received the electric shock then showed a painful expression to indicate that he was experiencing painful feelings. These confederates' performances provided a cue of how intergroup conflict was initiated. Participants from the Control Group were informed that each confederate performed the Stroop task on his own. After three trials, the confederate who performed worse than a standard (with 30% accuracy and reaction times shorter than 2000 ms or with 100% accuracy and reaction times shorter than 300 ms) would receive a painful or non-painful electric shock randomly given by a computer. One confederate illustrated receiving a painful shock and the other confederate illustrated receiving a non-painful shock. Phase 2 lasted for 15 min.

## Phase 3: viewing intergroup conflict and reporting punishment tendencies

Before being transported into the MRI scanner, the participant was informed that the two confederates in the test room (one ingroup member and one outgroup member, i.e., the Involved_Ingroup and Involved_Outgroup targets, respectively) would keep playing the game and the winner would decide whether to give the loser a painful or non-painful shock. The participant would be able to see a photo of the loser's face, indicating that he was experiencing painful or non-painful feelings, while inside the scanner. During four fMRI scans, a fixation was first presented with its duration varying among 2 s, 4 s, 6 s and 8 s on each trial (*Figure 1B*). An ID-photo of the Involved_Ingroup or Involved_Outgroup target was then presented for 2 s to indicate the person who lost the game. The participant had to judge whether the ID-photo showed an ingroup or an outgroup member by pressing one of two buttons on a response box (the relationship between left/right buttons and ingroup/outgroup members was counter-balanced across participants). Thereafter, the winner's choice, either a yellow circle to indicate a non-painful shock or a yellow lightning symbol to indicate a painful shock, was presented for 2 s. After a fixation with a duration varying among 2 s, 4 s, 6 s and 8 s, a photo of the loser's face was presented for 2 s to indicate being shocked (a photo with neutral expression indicated receiving a non-painful shock and a photo with painful expression indicated receiving a painful shock). The participant was asked to view the photo without any response. The ID-photos, lightning (and round) symbols and photos with expression, were subtended with a visual angle of 7.58° × 11.35°, 3.79° × 3.79°and 7.58° × 9.47° (width ×height) at a viewing distance of 80 cm. Each scan started with a 6 s fixation, and a task instruction was presented for 10 s followed by 16 trials. The procedure was programmed so that both the Involved_Ingroup and the Involved_Outgroup targets lost the game in half of the trials and, when losing the game, received painful shocks in half of the trials and non-painful shocks in the other trials. The trials in which a target received painful or non-painful shocks were presented in a random order.

The participant was also informed that, when Involved_Ingroup and Involved_Outgroup targets took a break during the competitive game, the participants had to perform a task to discriminate an ingroup member and an outgroup member who were not involved in the competitive game (Uninvolved_Ingroup and Uninvolved_Outgroup targets, respectively). In each trial, a fixation was first presented with its duration varying among 2 s, 4 s, 6 s and 8 s. An ID-photo of an Uninvolved_Ingroup or an Uninvolved_Outgroup target was then presented for 2 s. The participant had to judge whether the ID-photo showed an ingroup or an outgroup member by pressing one of two buttons on a response box. Thereafter, a photo of the target with neutral or painful expression was presented for 2 s (*Figure 1B*). The participant was asked to view the photo without any response. Similarly, there were four scans during which participants viewed Uninvolved_Ingroup and Uninvolved_Outgroup targets. Each scan started with a 6 s fixation, and task instruction was presented for 10 s followed by 16 trials. The procedure was programmed so that both Uninvolved_Ingroup and

Uninvolved_Outgroup targets showed painful expressions in half of the trials and showed neutral expressions in the other trials.

The scanning procedure was divided into two sessions. In each session, there were two scans when participants viewed Involved_Ingroup and Involved_Outgroup targets and two scans when participants viewed Uninvolved_Ingroup and Uninvolved_Outgroup targets. The order of the four scans in each session was counterbalanced across participants. After the first session, the participants were presented with photos of painful expressions of the four targets (Involved_Ingroup, Involved_Outgroup, Uninvolved_Ingroup, and Uninvolved_Outgroup) and rated their emotions for each target on a Likert Scale (1 = not at all, 9 = extremely strong) in response to the following questions: 'How painful do you think the target was?', 'How uncomfortable were you when viewing the target's pain?', 'How angry were you when viewing the target's pain?', 'How fearful were you when viewing the target's pain?', and 'How happy were you when viewing the target's pain?'. After the second session, the participants were presented with ID photos of the four targets and had to rate their attitudes toward the targets on a Likert Scale (1 = not at all, 9 = extremely) in response to the following questions: 'How much do you trust the target?' and 'How much do you like the target?'. The participants were also asked to report their punishment tendencies by selecting the intensity of an electric shock that they would like to apply to the loser on a Likert Scale (1 = not painful at all, 9 = intolerably painful). The order of the rating tasks was counter-balanced across participants. Phase 3 lasted for 60 min for each participant.

## Measures of endogenous OT

Participants were asked to not drink alcohol, caffeine, or medication within the 24 hr prior to their participation. Participants were asked to rinse their mouths with water immediately after lunch. Saliva was collected from each participant at three points in time. The first collection was conducted at the end of Phase 2 (e.g., after the introduction of intergroup conflict, Time-1). The second collection was conducted immediately after Phase three outside the scanner (i.e., after viewing intergroup conflict and reporting punishment tendencies during fMRI scanning, Time-3), and the third collection was conducted 15 min later (Time-3). Participants were asked to place a roll of cotton in their mouths and to chew on it for a minute until it became saturated. The roll of cotton was then placed in a Salivette (Sarstedt, Rommelsdorft, Germany). The samples were stored at −20°C until assayed. OT levels were assayed using a 96-plate commercial OT-ELISA kit (ADI-900-153A; Enzo Life Science). Measurements were performed in duplicate according to the kit's instructions, similar to the procedures used in previous studies (*van ljzendoorn et al., 2012*; *Bhandari et al., 2014*; *Tsuji et al., 2015*). The optical density of the samples and standards was measured at wavelengths of 405 nm, with correction between 570 nm and 590 nm. A four-parameter logistics curve fitting program was used for the calculation of the concentration of OT in the samples. Owing to the failure of OT measurements on a few participants, 37 and 40 participants were left in the Revenge and Control groups, respectively, for all further analyses related to OT levels.

## fMRI data acquisition and analysis

Brain images were acquired using a 3.0T GE Signa MR750 scanner (GE Healthcare; Waukesha, WI) with a standard eight channel head coil. Functional images were acquired by using T2-weighted, gradient-echo, echo-planar imaging (EPI) sequences sensitive to blood oxygenation level dependent (BOLD) signals ($64 \times 64 \times 32$ matrix with $3.75 \times 3.75 \times 5$ mm$^3$ spatial resolution, repetition time = 2000 ms, echo time = 30 ms, flip angle = 90°, field of view = $24 \times 24$ cm). A high-resolution T1-weighted structural image ($512 \times 512 \times 180$ matrix with a spatial resolution of $0.47 \times 0.47 \times 1.0$ mm$^3$, repetition time = 8.204 ms, echo time = 3.22 ms, flip angle = 12°) was acquired after the first four scans. Padded clamps were used to minimize head motion and earplugs were used to attenuate scanner noise. The stimuli were projected onto a screen at the head of the magnet bore using Presentation. Participants viewed the screen through a mirror attached to the head coil.

Functional images were preprocessed using SPM8 software (the Wellcome Trust Centre for Neuroimaging, London, UK). Head movements were corrected within each run and six movement parameters (translation; x, y, z and rotation; pitch, roll, yaw) were extracted for further analysis in the statistical model. The functional images were resampled to $3 \times 3 \times 3$ mm$^3$ voxels, normalized to the Montreal Neurological Institute (MNI) template space and then spatially smoothed using an isotropic

of 8 mm full-width half-maximum (FWHM) Gaussian kernel. Four participants from each group (i.e., Revenge and Control groups) were excluded from fMRI data analysis because their head movements exceeded 5 mm. Hence, 40 subjects in each group were included in further fMRI data analysis, which was identical for the Revenge and Control groups. In the first general linear model (GLM), brain activations were estimated using eight onset regressors to identify brain activations in response to painful vs. neutral expressions. These included 1) ID photos of Involved_Ingroup and Uninvolved_Ingroup targets; 2) ID photos of Involved_Outgroup and Uninvolved_Outgroup targets; 3) the symbol of painful shocks for involved targets, or the 2 s fixation for uninvolved targets; 4) the symbol of non-painful shocks for involved targets, or the 2 s fixation for uninvolved targets (Figure 1B); 5) photos of painful expressions of Involved_Ingroup and Uninvolved_Ingroup targets; 6) photos of neutral expressions of Involved_Ingroup and Uninvolved_Ingroup targets; 7) photos of painful expressions of Involved_Outgroup and Uninvolved_Outgroup targets; and 8) photos of neutral expressions of Involved_Outgroup and Uninvolved_Outgroup targets.

We conducted separate GLM analyses of involved targets (using the above onset regressors but distinguishing the symbols of painful/non-painful shocks for Involved_Ingroup and Involved_Outgroup targets) and uninvolved targets for the whole-brain regression analysis of the effect of endogenous OT and region-of-interest (ROI) analyses. The GLMs included the realignment parameters to account for any residual movement-related effect. The voxels showing significant event-related responses to painful vs. neutral expressions were created using a canonical haemodynamic response function (HRF). Here, our fMRI analyses locked BOLD responses to the onset of facial expressions rather than symbols of shock decisions because we focused on the relationship between OT-level and empathic neural responses to targets' pain. The onset of a symbol of a shock decision indicated what type of shocks (painful or nonpainful) the winner was going to give to the loser. The onset of a face with a painful or nonpainful expression indicated the time when the target started to receive a painful or nonpainful shock and an emotional (painful or nonpainful) response was initiated. A whole-brain random effect analysis was then conducted to reveal brain regions that showed reliable responses to painful vs. neutral expressions of all targets. Brain activations were defined using a voxel-level threshold of p<0.001, uncorrected and cluster-level threshold of p<0.05, FWE corrected.

## ROI analyses

ROI analyses were conducted to test (1) ingroup favoritism in neural responses to perceived pain, (2) group differences in the association between endogenous OT and mPFC activity in response to Involved_Ingroup target's pain, and (3) the correlation between mPFC activity in response to Involved_Ingroup target's pain and punishment propensity towards outgroup members. ROI analyses were also conducted in the mediation analyses that tested the mediation role of mPFC activity in response to an Involved_Ingroup target's pain in the association between endogenous OT and punishment tendency towards outgroup members. To define the coordinates of ROIs independently, a leave-one-out test in which whole-brain analyses of the contrast of painful vs. neutral expressions which collapsed all the targets was conducted using 79 of the 80 participants from the two subject groups using a combined voxel-level threshold of p<0.001, uncorrected and cluster-level threshold of p<0.05, FWE corrected. ROI coordinates were defined at the peak voxel the corresponding brain regions for the left-out participant. ROIs were then defined as a sphere with 5-mm-radius centered at the peak voxel of the seed regions for the left-out participant. The contrast values were extracted using MarsBaR (http://marsbar.sourceforge.net).

In the ROI analyses of the ingroup favoritism in neural responses to perceived pain, ROIs included the bilateral AI/IFG in the empathy network and the left TPJ and mPFC in the theory-of-mind network. The contrast values of these ROIs were extracted from the whole-brain analyses of Ingroup painful vs. neutral expression and Outgroup painful vs. neutral expression. In the ROI analyses of moderation and mediation analyses, the mPFC activity and left TPJ activity in response to Involved_Ingroup targets' pain was extracted from the whole-brain analysis of Involved_Ingroup target's painful vs. neutral expression.

## Moderation analysis

To examine whether intergroup conflict moderated the associations between endogenous OT at Time-1 and mPFC activity (or left TPJ activity) in response to Ingroup_In target's pain, we performed

moderated hierarchical regression analyses. To do this, we first defined ROIs using the leave-one-out method by calculating the contrast of painful vs. neutral expressions of one target by including 79 participants from the two subject groups. ROIs were defined as a sphere with 5-mm-radius centered at the peak coordinate of the mPFC or left TPJ. The contrast values of painful vs. neutral expressions were extracted using MarsBaR (http://marsbar.sourceforge.net) from the left-out participant. We then dummy coded the Group variable (i.e., Revenge and Control groups) as 0 and 1. The Group variable (the moderator), the contrast value of the ROI (the independent variable) and the OT level at Time-1 (the dependent variable) were entered into Hayes's PROCESS macro (Model 1) (*Hayes, 2017*). In addition, ingroup bias in closeness, emotions and attitudes were entered into the model as covariates. The moderator effect was indicated by a significant interaction effect between the moderator and the independent variable.

To examine whether involvement of the targets (e.g., Involved_Ingroup target vs. Uninvolved_Ingroup target) moderated the association between endogenous OT and mPFC activity in response to Ingroup_In target's pain, we performed another moderation analysis with a repeated measure as the moderator. To do this, we conducted a regression analysis with the endogenous OT level at Time-1 as the independent variable and the difference of the mPFC activities towards the Involved_Ingroup target and Uninvolved_Ingroup target as the dependent variable. The moderation effect was indicated if the independent variable significantly predicts the dependent variable. The analyses were performed using Montoya's MEMORE macro (Model 2, *Montoya, 2019*).

## Mediation analysis

We performed mediation analyses to examine whether the mPFC activity mediates the pathway from the endogenous OT to punishment tendency. To do this, we estimated four regression models: 1) whether the independent variable (OT) significantly accounts for the dependent variable (punishment tendency) when not considering the mediator (e.g., Path c'); 2) whether the independent variable (OT) significantly accounts for the variance of the presumed mediator (mPFC activity) (e.g., Path a); 3) whether the presumed mediator (mPFC activity) significantly accounts for the variance of the dependent variable (punishment tendency) when controlling the independent variable (OT) (e.g., Path b); and 4) whether the independent variable (OT) significantly accounts for the variance of the dependent variable (punishment tendency) when controlling the presumed mediator (mPFC activity) (e.g., Path c). To establish the mediation, the path c' is not required to be significant, and the only requirement is that the indirect effect a x b is significant. Given a significant indirect effect, if Path c is insignificant, the mediation is classified as indirect-only mediation, which is the strongest full mediation (*Kenny et al., 1998*; *Zhao et al., 2010*). A bootstrapping method was used to estimate the mediation effect. Bootstrapping is a nonparametric approach to estimate the effect-size and test the hypothesis that is increasingly recommended for many types of analyses, including mediation (*Shrout and Bolger, 2002*; *Mackinnon et al., 2004*). Rather than imposing questionable distributional assumptions, bootstrapping generates an empirical approximation of the sampling distribution of a statistic by repeated random resampling from the available data, and uses this distribution to calculate p-values and to construct confidence intervals. 5000 resamples were taken for our analyses. Moreover, this procedure supplies superior confidence intervals (CIs) that are bias-corrected and accelerated (*Preacher et al., 2007*; *Preacher and Hayes, 2008a*; *Preacher and Hayes, 2008b*). The analyses were performed using Hayes's PROCESS macro (Model 4, *Hayes, 2017*).

## Additional information

### Funding

| Funder | Grant reference number | Author |
|---|---|---|
| National Natural Science Foundation of China | 31661143039 | Shihui Han |
| National Natural Science Foundation of China | 31421003 | Shihui Han |
| National Natural Science Foundation of China | 31470986 | Shihui Han |

The funders had no role in study design, data collection and interpretation, or the decision to submit the work for publication.

## Author contributions
Xiaochun Han, Conceptualization, Data curation, Formal analysis, Validation, Investigation, Visualization, Methodology, Writing - original draft, Writing - review and editing; Michele J Gelfand, Conceptualization, Methodology, Writing - review and editing; Bing Wu, Ting Zhang, Wenxin Li, Tianyu Gao, Chenyu Pang, Taoyu Wu, Yuqing Zhou, Shuai Zhou, Data curation, Methodology, Writing - review and editing; Xinhuai Wu, Data curation, Supervision, Methodology, Writing - review and editing; Shihui Han, Conceptualization, Resources, Data curation, Formal analysis, Supervision, Funding acquisition, Validation, Investigation, Visualization, Methodology, Writing - original draft, Project administration, Writing - review and editing

## Author ORCIDs
Shihui Han (iD) https://orcid.org/0000-0003-3350-5104

## Ethics
Human subjects: Informed consent was obtained prior to the experiment. All participants were paid for their participation. This study was approved by the local ethics committee at the School of Psychological and Cognitive Sciences, Peking University.(#2015-12-04).

## Decision letter and Author response
Decision letter https://doi.org/10.7554/eLife.52014.sa1
Author response https://doi.org/10.7554/eLife.52014.sa2

# Additional files

## Supplementary files
• Source code 1. Scripts for the Bootstrap analysis of OT levels in *Figure 3B*.
• Source code 2. Scripts for the whole-brain analysis in *Figure 4A*.
• Source code 3. Scripts for the whole-brain analysis in *Figure 4B* _Ingroup.
• Source code 4. Scripts for the whole-brain analysis in *Figure 4B* _Outgroup.
• Source code 5. Scripts for the whole-brain regression analysis _Time 1.
• Source code 6. Scripts for the whole-brain regression analysis _Time 2.

• Supplementary file 1. Demographic information and psychological traits of the participants in the fMRI experiment. This file shows the means (SD) and statistics for comparisons between the Revenge and Control groups.

• Supplementary file 2. Results of group manipulation check. This file shows the means (and SD) of emotions and attitudes and statistics for comparisons between the Revenge and Control groups.

• Supplementary file 3. Factorial models of emotion and attitude rating items. This files shows the results of a factorial analysis that tested the discriminant validity of the eight items related to measures of emotions and attitudes. The analysis revealed two factors, which explained 62.60% of total variance. Factor one was the emotion factor (explaining 37.65% of variance), which included five items: empathy (0.653), unpleasant (0.906), anger (0.748), fear (0.837), and schadenfreude (–0.349). Factor 2 was the attitude factor (explaining 24.95% of variance), which included two items: likability (0.907) and trust (0.918).

• Supplementary file 4. Brain activations elicited by painful vs. neutral expressions across the Revenge and Control groups. This file shows the MNI coordinates of activated brain regions, cluster sizes, and Z values.

• Supplementary file 5. Brain activations elicited by painful vs. neutral expressions in the Revenge group. This file shows the MNI coordinates of activated brain regions, cluster sizes, and Z values.

- Supplementary file 6. Brain activations elicited by painful vs. neutral expressions in the Control group. This file shows the MNI coordinates of activated brain regions, cluster sizes, and Z values.

- Supplementary file 7. The results of the moderation analysis. This file shows the statistical details of the moderation analysis that examined how group identity (Revenge vs. Control group) moderated the relationship between endogenous OT (Time-1) and mPFC activity in response to Involved_Ingroup target's pain.

- Supplementary file 8. The results of the mediation analysis. This file shows the statistical details of the moderation analysis that examined whether the mPFC activity mediated the relationship between endogenous OT (Time-1) and punishment tendencies towards the Involved_Outgroup target.

- Supplementary file 9. The results of the mediation analysis. This file shows the statistical details of the moderation analysis that examined whether the mPFC activity mediated the relationship between endogenous OT (Time-1) and punishment tendencies towards the Uninvolved_Outgroup target.

- Supplementary file 10. Demographic information and psychological traits of the participants in the new behavioral experiment. This file shows the means (SD) and statistics for comparisons between the Revenge and Control groups.

- Supplementary file 11. Ingroup favoritism in self-report of emotions, attitudes, punishment tendencies, and punishment decisions in the new behavioral experiment. This file shows the means (SD) and statistics for comparisons between the Revenge and Control groups.

- Transparent reporting form

## Data availability

All data generated or analysed for figures of this study are included in the manuscript and supporting files. Source data files have been provided for Figures 2–6.

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
