## [Decision Letter]

**Acceptance summary:**

This paper reports a well-designed study investigating the neural basis of revenge during intergroup conflict. Through its clever design that includes multiple control conditions, the study is able to convincingly demonstrate behavioral, hormonal (endogenous oxytocin), and neural measures specifically related to viewing the pain of in-group members inflicted by out-group members. Both the neuroimaging and the endogenous oxytocin results are novel contributions in themselves, and the combination is unique.

**Decision letter after peer review:**

Thank you for submitting your article "A neurobiological association of revenge propensity during intergroup conflict" for consideration by *eLife*. Your article has been reviewed by three peer reviewers, one of whom is a member of our Board of Reviewing Editors, and the evaluation has been overseen by Floris de Lange as the Senior Editor. The following individual involved in review of your submission has agreed to reveal their identity: Carsten De Dreu (Reviewer #2).

The reviewers have discussed the reviews with one another and the Reviewing Editor has drafted this decision to help you prepare a revised submission.

Summary:

This is a well-designed study investigating neural mechanisms of revenge propensity. The study includes a manipulation to induce in-group/out-group attitudes and subsequently measures fMRI responses to images of painful faces depicting in- and out-group members, with (participants are made to belief) pain inflicted to in-group members by either an out-group member or a computer (control group). The study additionally includes pre-post measures of endogenous oxytocin. The study thus includes several different dependent measures (behavioral, neural, hormonal) and many experimental variables. The elaborate design allows for controlling factors of non-interest. The key result is that humans respond stronger -- behaviorally, hormonally, and neurally -- to punishment of in-group compared to out-group members, that such stronger responding can be best typified as empathy-based, and that such empathy-for-ingroup-pain predict retaliatory responding to out-group members. Overall, the paper reports an interesting set of results, bringing together insights from various literatures on (revenge propensity during) intergroup conflict. Both the fMRI and the endogenous oxytocin results are novel contributions in themselves, and the combination is unique.

Essential revisions:

1) Having such an elaborate design creates a challenge for conducting and reporting the analyses. These complexities made it difficult to digest what exactly was done when, why and how, and what analyses were performed (and why). There are so many possible ways to analyze the data that this needs to be done in a very principled manner, with each analysis well-motivated (related to main hypotheses) and including the relevant control conditions. There is a risk of selectively reporting only the significant results and it is easy to lose the reader in the many variables, interactions, mediation models etc. The paper would benefit from relaxing the write-up and some re-organization of (grouping together) analyses and results. One possibility is to report liking ratings and behavioral revenge measures first, with a pointed summary. Subsequently all analyses pertaining to oxytocin. Then all analyses pertaining to neuro-imaging. Only then analyses pertaining to the linkages between revenge and oxytocin, and oxytocin and neural activity. Other, even better, organization may be possible.

The fMRI reporting moves back-and-forth between whole-brain analyses and ROI-based contrasts, and results are difficult to digest. Again, the writing may be relaxed a bit to "take the reader by the hand" and interim summary statements may help.

2) The link between the specific hypotheses, reported analyses, and the theoretical background should be clarified and returned to in the Results section when motivating the analyses. The Introduction contrasts possible roles of the empathy network and the theory-of-mind network, but the hypotheses are not specified at that level (only mentioning "brain activity"); the analyses similarly don't compare the involvement of these networks directly. The whole brain contrasts to painful>neutral expressions, revealing both the empathy and theory-of-mind networks would seem to be of central importance but is now presented as a supplementary figure. These regions could possibly be used as ROIs to test how these two networks are modulated by the manipulated variables.

3) The OT results differ greatly between groups, already at T1 where there should be no group difference (if I understood the design correctly). It looks like the OT measures for the Revenge group are all doubled relative to the Control group. This makes any subsequent analysis involving OT group comparisons (including some of the key results of the paper) problematic, also because the variability in the Control group at T1 is much smaller than in the Revenge group. Why does this not invalidate the analyses involving OT measures?

Furthermore, it is quite surprising that all correlations/mediations are based on OT in time 1. Should this not be with OT (time 2 minus time 1) which reflects ingroup pain induced OT change? For the main effect, the authors used Time 2 minus Time 1 to support "intergroup conflict encountered by Revenge group is associated with increased salivary level of oxytocin". Later, the authors only used results from Time 1 to support "The medial prefrontal activity to ingroup pain mediates the association between endogenous oxytocin and propensity to seek revenge by giving painful electric shocks to outgroup". It is post hoc and not logical. Why it is baseline OT levels rather than changes in OT levels that predict mPFC activity?

---

## [Author Response]

Essential revisions:1) Having such an elaborate design creates a challenge for conducting and reporting the analyses. These complexities made it difficult to digest what exactly was done when, why and how, and what analyses were performed (and why). There are so many possible ways to analyze the data that this needs to be done in a very principled manner, with each analysis well-motivated (related to main hypotheses) and including the relevant control conditions. There is a risk of selectively reporting only the significant results and it is easy to lose the reader in the many variables, interactions, mediation models etc. The paper would benefit from relaxing the write-up and some re-organization of (grouping together) analyses and results. One possibility is to report liking ratings and behavioral revenge measures first, with a pointed summary. Subsequently all analyses pertaining to oxytocin. Then all analyses pertaining to neuro-imaging. Only then analyses pertaining to the linkages between revenge and oxytocin, and oxytocin and neural activity. Other, even better, organization may be possible.The fMRI reporting moves back-and-forth between whole-brain analyses and ROI-based contrasts, and results are difficult to digest. Again, the writing may be relaxed a bit to "take the reader by the hand" and interim summary statements may help.

We appreciate your point and thank you for your suggestions for how to reorganize the Results section. Accordingly, we have modified Results section to present the results in the following order:

1) Results of survey ratings of attitudes and emotions and revenge propensity measures

2) Results of oxytocin measures

3) fMRI results of brain responses to perceived pain (*Note*: we also further justified why we use whole-brain and ROI analyses to address our research questions)

4) Results of the relationship between the oxytocin measures and brain activity in response to perceived pain (i.e., mPFC activity)

5) Results of the association between mPFC activity to ingroup pain and revenge propensity

6) Results of the mediation analyses (mPFC activity mediation of the relationship between oxytocin and revenge propensity)

In order to further help the flow of the manuscript, we also motivate why we organized the Results section in this way in the introduction. For further clarity, at the beginning of each subsection of Results we explained what questions and hypotheses we sought to test and how the analyses reported tested these hypotheses. Finally, we also presented a short summary at the end of each subsection of Results to highlight key findings and implications. We believe that these modifications will help readers to understand the results in a more clear and coherent manner.

2) The link between the specific hypotheses, reported analyses, and the theoretical background should be clarified and returned to in the Results section when motivating the analyses. The Introduction contrasts possible roles of the empathy network and the theory-of-mind network, but the hypotheses are not specified at that level (only mentioning "brain activity"); the analyses similarly don't compare the involvement of these networks directly. The whole brain contrasts to painful>neutral expressions, revealing both the empathy and theory-of-mind networks would seem to be of central importance but is now presented as a supplementary figure. These regions could possibly be used as ROIs to test how these two networks are modulated by the manipulated variables.

Thanks for these important suggestions. We have now made the theoretical background clear for each specific hypothesis before reporting our analyses and results. We are also specific now in the Introduction regarding our hypotheses that empathy and theory-of mind networks are associated with endogenous OT. As we discuss in the Introduction, OT receptors have been observed in both networks and additional research has illustrated that nasal administration of OT affected activities in both networks. These findings suggest that the brain activity in both networks in response to ingroup members' pain may be associated with endogenous OT. We conducted whole-brain regression analyses using salivary levels of OT as predictors to examine the association between endogenous OT and brain responses to perceived ingroup pain so as not to bias either the empathy network or the theory-of-mind network. This is clarified on in subsection “Endogenous OT predicts mPFC activity in response to ingroup pain”.

We moved the results of whole-brain analyses of neural responses to perceived pain into the main text, and as suggested by the reviewers, we conducted ROI analyses using the leave-one-out method to define ROI (to avoid double-dipping) to examine which (the empathy or theory-of-mind) network is modulated by ingroup/outgroup manipulation and by Revenge. These results are reported in the revision (subsection “Brain responses to perceived pain in Revenge and Control groups”). We believe these changes have significantly improved the manuscript.

Our work used whole-brain analyses to examine which brain regions are activated in response to ingroup pain, and as noted, we showed evidence that both the empathy and theory-of-mind networks were activated. However, we did not focus on which network was more strongly activated in response to ingroup pain because there are a number of issues that make it difficult to address this question appropriately. For example, there is the question of how to define ROIs that do not bias activities in different networks, how large an ROI should be for different nodes of a network and of different networks, among other issues. The conventional ROI analysis compares activity in an ROI between two different conditions rather than to compare activities in two different ROIs in the same condition. This is why we did not compare the activity of the two networks directly.

3) The OT results differ greatly between groups, already at T1 where there should be no group difference (if I understood the design correctly). It looks like the OT measures for the Revenge group are all doubled relative to the Control group. This makes any subsequent analysis involving OT group comparisons (including some of the key results of the paper) problematic, also because the variability in the Control group at T1 is much smaller than in the Revenge group. Why does this not invalidate the analyses involving OT measures? Furthermore, it is quite surprising that all correlations/mediations are based on OT in time 1. Should this not be with OT (time 2 minus time 1) which reflects ingroup pain induced OT change? For the main effect, the authors used Time 2 minus Time 1 to support "intergroup conflict encountered by Revenge group is associated with increased salivary level of oxytocin". Later, the authors only used results from Time 1 to support "The medial prefrontal activity to ingroup pain mediates the association between endogenous oxytocin and propensity to seek revenge by giving painful electric shocks to outgroup". It is post hoc and not logical. Why it is baseline OT levels rather than changes in OT levels that predict mPFC activity?

We thank reviewers for these very important comments. We believe that there are three questions here:

Regarding Question 1(Why does OT-level at Time 1 differ between Revenge and Control groups?), we must clarify the timing of salivary collection in our experimental design. Saliva was collected at Time 1 *after* introducing the conflict situation to participants of the Revenge group. Importantly, we've made clear in the revision that the conflict situation was different between the Revenge and Control groups at Time 1. A participant in the Revenge group had observed an ingroup member and an outgroup member who competed with each other and applied painful/non-painful electric shocks to each other. By contrast, a participant in the Control group had observed an ingroup member and an outgroup member who play the game independently and received painful/non-painful electric shocks from a computer. Therefore, at Time 1, the Revenge and Control groups already had different experiences of intergroup conflict which enabled us to directly test the hypothesis that early experiences of intergroup conflict causes a rise in OT level at Time 1 as compared to a control condition. We apologize that we had not made this point more clear in the original submission, and we believe that Question 1 was asked because there was a misunderstanding that the Revenge and Control groups had the same experience at Time 1 (which they did not). We now make this very clear in the revision in subsection “Punishment tendencies in Revenge and Control groups” and “Measures of endogenous OT”. We reviewed critical research that has shown that urine oxytocin level was escalated immediately after one group of chimpanzees had encountered another group of chimpanzees (Samuni et al., 2017). Accordingly, our finding of OT-level difference at Time 1 between the Revenge and Control groups was part of our design and fully expected, and consistent with the finding of the chimpanzees (Samuni et al., 2017). More generally, the results of both chimpanzee and our study suggest that endogenous oxytocin may increase as early as the initial experience of group conflict (e.g., encountering outgroup in chimpanzees in Samuni et al. and viewing an outgroup member brought physical harm to an ingroup member in our work). We include additional discussion regarding Question 1 in the revised Discussion.

Regarding Question 2 (Why was the variability in OT-level at T1 smaller in the Control group than in the Revenge group?), we agree that it is critical to clarify whether the results of OT measures reflect our experimental manipulation or a technical problem with the OT analyses. To address this question, we analyzed the variability of OT-levels at three time points and across the two groups. This is critical because Question 2 and the related concern of our OT measures/analyses are based on the assumption that “the variability in OT-level at T1 smaller in the Control group than in the Revenge group”. Therefore, we conducted three nonparametric Levene tests for equal variances (Nordstokke, D. W., and Zumbo, B. D. (2010). A new nonparametric Levene test for equal variances. Psicológica, 31(2), 401-430.) to statistically examine the differences in standard deviation (SD) of OT levels at Time 1, 2 and 3 between the Revenge and Controls groups. The results showed that there was no significant difference in SD of OT levels between Revenge and Control groups (Time 1/Time2/Time3: F(1,75) = 2.53, 1.75, and 1.37, p = 0.116, 0.190, and 0.245). Thus the results indicate no evidence for the assumption that the variability of OT levels in the Control group were smaller than that in the Revenge group. Accordingly, we believe that the variability of the OT-level cannot invalidate our analyses involving OT measures. The impression that the variability in the Control group at T1 is much smaller than in the Revenge group was produced possibly because we did not make clear the meaning of Figure 2A, in which the vertical lines illustrate the maximum and minimum values rather than SD. We apologize for this. In the revision we modified Figure 3 so as to make clear the SD in different conditions by plotting the results in a different way. We hope that the new figure does not make confusions regarding the SD.

Regarding Question 3 (Why it is baseline OT levels rather than changes in OT levels that predict mPFC activity?), we should first clarify that OT level at Time 1 is not a baseline because the participants in the Revenge group had the initial experience of witnessing intergroup conflict at Time 1 whereas the Control group did not. To test the association between brain activity in response to ingroup pain and OT levels at Time 1 (after the initial experience of witnessing intergroup conflict) and Time 2 (after the whole fMRI scanning procedure) have different meanings. As we noted; "If the association between endogenous OT to brain responses to ingroup pain serves as a neurobiological correlate of revenge propensity during intergroup conflict, endogenous OT after the initial intergroup conflict at Time 1 should predict subsequent brain responses to ingroup pain which may then further predict revenge propensity. Accordingly, we first conducted a whole-brain regression analysis to examine whether OT levels at Time 1 predicts the brain responses to perceived ingroup pain. As discussed below, this analysis revealed an association between the mPFC activity and OT level at Time 1 in the Revenge group. In order to then estimate whether the OT-mPFC association was specific to OT levels at time 1, we conducted a second whole-brain regression analysis to examine brain responses to ingroup pain that were associated with endogenous OT measured after fMRI scanning at Time 2." The results of our whole-brain regression and ROI-based moderation analyses suggest that intergroup conflict enhanced the link between endogenous OT at Time 1 and mPFC activity in response to ingroup pain. As we note, we also tested associations between OT level at Time 2 and brain responses to ingroup pain, but the results provided no evidence for revenge specific association between brain responses to ingroup pain and further changes of endogenous OT at Time 2. These results provide bases for further examination of the functional role of the mPFC activity in mediating the association between endogenous OT after initial experience and revenge propensity. We hope we have made these clear in the revision.